# What Is There to Buy? An Analysis of the Food Environment in Public and Private Schools in the Federal District

**DOI:** 10.3390/ijerph22091331

**Published:** 2025-08-26

**Authors:** Giovanna Soutinho Araújo, Vivian S. S. Gonçalves, Ariene Silva do Carmo, Maurício T. L. de Vasconcellos, Natacha Toral

**Affiliations:** 1Postgraduate Program in Human Nutrition, Faculty of Health Sciences, Center for Epidemiological Studies in Health and Nutrition—NESNUT, University of Brasília (UnB), Brasília 70910-900, DF, Brazil; arienecarmo@gmail.com (A.S.d.C.); natachatoral@unb.br (N.T.); 2Department of Biological and Health Sciences, Universidade do Distrito Federal Professor Jorge Amaury Maia Nunes (UnDF), Brasília 71503-502, DF, Brazil; 3Graduate Program in Public Health, Faculty of Health Sciences, University of Brasília (UnB), Brasília 70910-900, DF, Brazil; vivian.goncalves@unb.br; 4Sociedade para o Desenvolvimento da Pesquisa Científica—SCIENCE, Rio de Janeiro 20231-050, RJ, Brazil; mautlv@gmail.com

**Keywords:** food environment, school food environment, food retail, healthy eating, ultra-processed foods

## Abstract

This descriptive ecological study evaluated the food environment of 18 public and private schools in the Federal District (DF), Brazil, by analyzing food availability within schools and in their surroundings (250 m, 400 m, and 800 m buffers). Food retail outlets (FROs) were georeferenced and classified according to the NOVA food classification. School principals were interviewed to assess the in-school food environment. Analyses considered the Social Vulnerability Index (SVI; low or medium/high) and school type. Among 911 FROs identified, 40.2% predominantly sold ultra-processed products. Most schools (83.3% within a 250 m radius) had at least one nearby FRO. Private schools—mostly in low-SVI areas—had higher densities of surrounding FROs at all buffer distances, with significance for total density at 400 m (*p* = 0.03) and for unhealthy outlets at 800 m (*p* < 0.01). Low-SVI areas had higher densities of both healthy (*p* = 0.01) and unhealthy (*p* < 0.01) outlets, with differences across multiple buffers. In canteens, sugar-sweetened beverages were the most common ultra-processed items (75%). The median ratio of ultra-processed to minimally processed food subgroups was 2.7 (0.5–6.0), and all private schools with a canteen sold at least one item prohibited by current regulations. Overall, the DF school food environment was characterized by a predominance of unhealthy foods, with disparities by school type and social vulnerability.

## 1. Introduction

Schools play a crucial role in shaping the eating habits of children and adolescents. Students spend many hours in this environment and often consume a significant portion of their daily caloric intake while at school [1,2]. They frequently make food choices without adult supervision [3].

Beyond their direct influence on food consumption, schools are key social spaces that help shape long-term dietary behaviors [4], as students interact, learn to manage emotions, and develop health-related habits within this environment. Both the internal facilities and the surrounding environment influence the adoption of healthy eating practices and physical activity [5].

The Food and Agriculture Organization (FAO) [6] defines the “school food environment” as all spaces, structures, and situations within and around the school where food is available, sold, or consumed. This definition includes the nutritional quality of foods, their prices, the availability of food and nutrition information, and marketing practices [6].

Several studies have examined how these factors manifest in Brazil. For instance, research shows that a greater variety of ultra-processed foods in school canteens is linked to a more frequent consumption of such products by students [7,8,9,10]. Despite the growing number of studies on food environments in the country, methodological inconsistencies limit the comparability and applicability of findings, such as variations in defining school surroundings, classifying food outlets, and assessing internal environments [11,12]. Moreover, most research has been concentrated in Southeastern Brazil, resulting in a lack of data for other regions. This imbalance highlights the need for context-specific evidence to support public policies tailored to local realities [11].

This gap is especially relevant in the Federal District (DF), located in Central-Western Brazil and home to the nation’s capital. The DF is characterized by marked inequalities across its administrative areas, making it one of the most income-segregated regions in the country [13,14]. These socioeconomic disparities, in turn, shape food environments across its territories [15,16]. Understanding how food availability and access within and around schools are configured is therefore essential for developing effective strategies to promote the health of children and adolescents.

To address this gap, this study investigates the following research question: How is the food environment configured within and around public and private schools in the DF, and how is this configuration associated with social vulnerability? In this study, social vulnerability refers to the susceptibility of groups to adverse impacts on health and well-being resulting from multidimensional social and territorial inequalities, including differences in income, education, quality of life, and access to services and other resources [17]. We mapped the surroundings and characterized the internal food environment of public and private schools in the DF, analyzing them based on school type and the degree of social vulnerability in each region. This study provides novel, locally grounded evidence to guide more effective and context-sensitive school food policies in Brazil.

The remainder of this article is organized as follows. Section 2 presents the methods, including the study design and context, sample calculation and selection, the characterization of the school food environment, and data processing and analysis. Section 3 contains the results, covering the characterization of the school surroundings and the internal and surrounding food environments based on school staff reports. Section 4 discusses the findings in light of the existing literature and highlights policy implications. Section 5 concludes the study with key messages and recommendations.

## 2. Methods

### 2.1. Study Design and Context

This study was part of a larger project entitled “School food environment in the Federal District (AMBIAS): association with obesity, food choices, and perceptions of adolescents.” It was approved by the Research Ethics Committee of the Faculty of Health Sciences of the University of Brasília (protocol CAAE No. 17780819.4.0000.0030) and authorized by the School for the Improvement of Education Professionals of the Federal District Department of Education.

The DF, located in the Central-Western region of Brazil, has 33 administrative regions (ARs) and a population of 2,817,381 [18]. Its average per capita income is BRL 3357.00, with significant inequality between the ARs. Although it has a very high Human Development Index (0.824) [19], its Social Vulnerability Index (SVI) was 0.33 in 2021—the most recent year with available data—indicating medium vulnerability and reflecting intermediate levels of disadvantage in access to services, opportunities, and quality of life [14]. The SVI ranges from 0 to 1, with higher values indicating greater vulnerability, and is categorized into five levels (very low to very high).

This descriptive ecological study included urban public and private schools in the DF offering ninth-grade classes. Schools were selected using probabilistic sampling stratified by school type. The study population comprised all 356 urban schools with ninth-grade classes (159 public and 187 private) [20]. The inclusion followed that of the Brazilian National Survey of School Health (PeNSE—Pesquisa Nacional de Saúde do Escolar) [20], a nationwide survey conducted by the Brazilian Institute of Geography and Statistics (IBGE) in partnership with the Ministry of Health and supported by the Ministry of Education. PeNSE monitors risk and protective factors for school health, providing updated national estimates for the target population. Aligning the inclusion criteria with PeNSE allows us to make a comparison with national estimates [21].

### 2.2. Sample Calculation and Selection

The sample size was calculated to estimate a minimum proportion of 3%, with a relative error of 6% at the 95% confidence level (or 5% significance level). Based on these parameters and assuming a sample design effect of 2.6, a sample of 20 schools (9 public and 11 private) was obtained. However, two private schools refused to participate, resulting in a final sample of 18 schools (9 public and 9 private), providing an equal allocation between the types. The reasons for refusal were not related to school characteristics. In most probability samples, when no information on non-respondents is unavailable, it is commonly assumed that no systematic differences exist between participants and non-participants. This assumption minimizes the risk of selection bias and supports the validity of the sample design [22].

Although the small sample size (18 schools) may limit the statistical precision of estimates, the probabilistic design ensures that the sample is representative of the target population.

### 2.3. Characterization of School Food Environment

Based on the conceptual model proposed by Castro and Canella [23], the food environment was assessed in two modules: (1) commercial establishments surrounding schools and (2) food available inside schools.

#### 2.3.1. Characterization of School Surroundings: Data Sources and Classification Methods

Private food retail establishments in school surroundings were identified and classified in three steps: (1) extraction from administrative records; (2) verification via Google Street View (virtual audit); and (3) classification based on predominant food sales. This multi-step procedure minimized the risk of omitting eligible establishments and improved classification accuracy, thereby addressing limitations of secondary administrative data in food environment studies [11,24]. The methodological approach is consistent with previous studies conducted in Brazil and abroad [15,25,26], making this study comparable with research that combines administrative data and observational verification to characterize school and community food environments.

A database containing addresses of schools and commercial establishments selling food for immediate consumption was prepared using twelve codes from the National Classification of Economic Activities (CNAE—Classificação Nacional de Atividades Econômicas) [27]. Establishments matching these codes were extracted from the 2021 Annual List of Social Information (RAIS—Relação Anual de Informações Sociais) [28], the most recent available at the time of data collection (May 2023). The RAIS database, also used in the 2018 CAISAN study [29], was selected as it represents the most comprehensive public source of labor activity data in Brazil.

Georeferenced school data were obtained from the DF Department of Education (2022) and the Geoportal-DF of the State Department of Urban Development and Housing [30], which also includes geospatial data on street fairs.

Establishments identified through RAIS were subsequently audited virtually using Google Street View (2023). This process involved verifying the existence, geographic location, and building facade of each listed establishment. When discrepancies were identified between the registered CNAE code and the actual business activity visible in the images, corrections were made. Additionally, eligible establishments not originally listed were also identified and included. Universal Transverse Mercator coordinates and corresponding zones were recorded for all verified establishments.

The classification of establishments followed the NOVA food classification system [31], which groups foods into the following: (i) unprocessed or minimally processed foods (MPFs): fresh or minimally processed foods and culinary preparations based on them; (ii) processed culinary ingredients: products extracted from MPFs or nature; (iii) processed foods: MPFs with added salt, sugar or oil, usually with two or three ingredients; and (iv) ultra-processed foods (UPFs): industrial formulations made mostly or entirely from substances derived from foods and additives, with little or no intact MPFs, and often ready to consume.

Following the methodology proposed by the Brazilian Ministry of Development and Social Assistance, Family and Fight against Hunger (MDS—Ministério do Desenvolvimento e Assistência Social, Família e Combate à Fome) [32], consumer purchase data from the 2017/2018 Brazilian Household Budget Survey [33] were used to determine the predominant sales profile for each type of establishment, stratified by Federative Unit. This classification reflects the average consumer purchase profile rather than the store’s intended product offering.

Based on this information and guided by the Brazilian Dietary Guidelines [34], establishments were grouped into five categories: (G1) fresh: establishments with 50% or more sales of MPFs (≥50% MPF sales); (G2) mixed fresh: with at least 40% sales of MPF and processed foods and less than 20% sales of UPF (≥40% MPF sales and <20% UPF sales); (G3) mixed processed: establishments with sales of at least 40% UPF and less than 20% MPF (≥40% UPF sales and <20% MPF) or establishments with at least 70% of UPF and processed food sales and less than 20% of MPF sales (≥70% UPF and processed foods with <20% MPF); (G4) ultra-processed: 50% or more UPF sales (≥50% UPF sales); and (G5) Other mixed: establishments not falling into the previous categories.

In the Federal District, no establishments were classified as G2 (mixed fresh). As noted in the original methodology, the distribution of establishments across categories may vary by state. By type, G1 included supermarkets, green grocers and grocery stores, as well as street fairs; G3 comprised bakeries and confectioneries; G4 included retailers of sweets, snack bars and similar, convenience stores and bars; and G5 included hypermarkets, general food retailers, restaurants and similar, and street food services. Fishmongers and butchers were excluded, as they typically do not offer ready-to-eat food and are not commonly frequented by adolescents. School canteens were also excluded to avoid duplication.

For analysis, establishments were grouped into “Healthy” (G1 and G5) and “Unhealthy” (G3 and G4), according to their predominant product profile. From the georeferenced data, Euclidean buffers of 250 m, 400 m, and 800 m were created around each school using QGIS 3.40, identifying food establishments within those distances. These distances correspond to approximately 3, 5, and 10 min walks, considering average adolescent walking speed (4–5 km/h) [35,36]. In cases where buffers from different schools overlapped, some establishments were counted more than once, as the analysis was conducted separately for each school’s surroundings. The final dataset included establishment counts and proportions by category within the buffers, as well as density per 10,000 inhabitants.

#### 2.3.2. Characterization of the School Food Environment: Data Collection from School Staff

A questionnaire was applied through interviews with the principal or pedagogical coordinator of each school. The instrument investigated food marketed in the school and its surroundings, the performance of Food and Nutrition Education (FNE) activities, and structural aspects of the food environment. These included: presence of a canteen and/or restaurant; how food is supplied to students (free of charge or purchased at the canteen and/or restaurant); availability of microwaves for students; presence of a nutritionist; and permission for snacks from outside the school. Questions also focused on where meals were eaten—cafeteria, canteen area, or classroom—and the time available for meals, categorized as 10–15 min, 20 min, or at least 30 min. The questionnaire also inquired about the presence of alternative points of sale for food and/or beverages at the school entrance or nearby.

Food sold at the school was selected from a list of 17 items often sold in these establishments [21,37], including the following: (1) sugar-sweetened beverages (box juice, powdered juice, chocolate drinks, teas, energy drinks, and sports drinks); (2) filled and unfilled cookies; (3) industrialized cakes, chocolates, candy, sweets, lollipops, and gum; (4) Brigadeiro; (5) cupcakes; (6) fresh fruits, natural fruit juice, and pulp juice; (7) gelatin; (8) hamburgers; (9) instant noodles; (10) popsicles and ice cream; (11) pizza; (12) soft drinks; (13) chips, microwave popcorn, and industrialized sweet popcorn; (14) savory pastries (with meat, chicken, and/or cheese); (15) savory pastries (with ham, pepperoni, hamburger meat, and/or sausage); (16) tapioca and couscous; and (17) others.

For alternative points of sale (street vendors) and commerce in the surroundings, a list was presented with the same items already mentioned, in addition to fresh popcorn, boxed lunches (with full meals), and churros, totaling 20 items [21,37] (Table 1).

The following indicators, as proposed by Tavares et al. [38], were used to assess the healthiness of the canteens and food stores around schools:(1)Prop-MPF: The proportion of fresh, minimally processed, or processed food subgroup availability among all MPF subgroups (1):(1)$$Prop\text{-}\mathrm{MPF}=\frac{Number\;of\;MPF\;subgroups\;sold}{Total\;MPF\;subgroups\;selected}\times 100$$

(2)Prop-UPF: The proportion of the availability of ultra-processed food subgroups among all UPF subgroups (2):


(2)
$$Prop\text{-}\mathrm{UPF}=\frac{Number\;of\;UPF\;subgroups\;sold}{Total\;selected\;UPF\;subgroups}\times 100$$


(3)UPF/MPF Ratio: The ratio between UPF availability and MPF availability, expressing the relative predominance of UPF over MPF (3):


(3)
$$UPF/MPF\;\mathrm{Ratio}=\frac{Total\;UPF\;items\;sold\;}{Total\;MPF\;items\;sold}$$


Values greater than 1 indicate a higher commercialization of UPF, while values less than 1 indicate a higher commercialization of MPF.

(4)Healthiness Index (HI): Considers the availability of food in the MPF subgroup and the non-availability of food in the UPF subgroup, assigning one point for each MPF subgroup available at school and one point for each UPF subgroup not available. Missing MPF subgroups or present UPF subgroups are not scored. The final score is calculated according to Equation (4):


(4)
$$\mathrm{Healthiness}\;\mathrm{Index}\;(\mathrm{HI})=\frac{\left(Total\;score\;of\;MPF\;subgroups\;available+Total\;score\;of\;UPF\;subgroups\;not\;available\right)}{\;Total\;number\;of\;subgroups\;evaluated}\times 100$$


The score ranges from 0 to 100, with higher values indicating greater healthiness of the establishment.

Foods not marketed by any school were excluded from the HI calculation to avoid distortions. Similar foods were grouped into categories according to the level of processing, such as sugar-sweetened beverages and energy drinks; industrialized cakes, brigadeiro, and cupcakes; and popsicles, ice cream, and açaí.

Indicator calculation for school canteens was based on 13 subgroups: three MPFs (fresh fruit, natural fruit juice, and pulp juice; savory pastries with meat, chicken, and/or cheese; and tapioca and couscous) and ten UPFs (sugar-sweetened beverages; filled/unfilled cookies; sweets; hamburger; popsicle, ice cream, and açaí; pizza; soft drink; chips and popcorn; savory pastries with sausages; and gelatin). Instant noodles were excluded because they were not sold in any school (Table 1).

For school surroundings, the following were excluded because they were not offered: instant noodles, fruit, cupcakes, gelatin, and energy drinks. Churros were classified as UPF, while boxed lunch (full meal) and popcorn were classified as MPFs, totaling 14 subgroups.

No public school in the study had a canteen and/or restaurant. Since 2019, all commercial canteens have been removed from public schools in the Federal District [39]. Thus, only private school canteens were analyzed, except for one school without a canteen (*n* = 8).

### 2.4. Data Processing and Analysis

The descriptive analysis included a calculation of absolute and relative frequencies for categorical variables and medians and interquartile ranges or minimum and maximum values for quantitative variables, given the asymmetric data distribution. Analyses were stratified according to the SVI of the region where the school was located and the type of school (public or private).

The SVI was classified as low (<0.300), medium (0.300–0.399), or high (>0.400) [14]. For this study, the medium and high categories were combined into a single group (“medium/high vulnerability”) to contrast areas with greater and lower social vulnerability, thereby highlighting disparities in food access and regions at higher nutritional risk [40,41,42].

Analyses of establishment density in the school environment considered the 250 m, 400 m, and 800 m buffers and the categories of healthy and unhealthy establishments. The spatial distribution of the establishments is represented in a figure created using QGIS software, version 3.40.

Median comparisons employed the Mann–Whitney test, and proportion comparisons used the chi-square or Fisher’s exact test, with a 95% confidence interval. All analyses were conducted in Stata software, version 16.11, with a significance level of 5%.

## 3. Results

### 3.1. Characterization of School Surroundings

In the surroundings of the eighteen analyzed schools (nine public and nine private), 911 food establishments were identified within the 800 m buffer. Of these, 17.5% were in G1 (MPF), 9.2% in G3 (mixed processed), 40.2% in G4 (UPF), and 33.1% in G5 (other mixed). Overall, 50.6% were classified as healthy and 49.4% as unhealthy.

By school type, the total number of food retail outlets within the 800 m buffer was 948, with 71.0% located around private schools and 29.0% around public schools. These total amounts included repeated establishments due to buffer overlap, particularly in areas where schools were in close proximity. In the 400 m buffer, 60.9% of outlets were located around private schools and 39.1% around public schools, while in the 250 m buffer, the proportions were 53.4% and 46.6%, respectively. The difference between public and private schools was statistically significant in the 800 m buffer (*p* < 0.001), whereas no significant differences were observed for the 400 m and 250 m buffers.

Within the 250 m buffer, 38.9% of schools (*n* = 7) had no healthy establishments, while 77.8% (*n* = 14) had unhealthy establishments. Three schools (16.7%) had no establishment in this radius. In the 400 m buffer, 11.1% (*n* = 2) lacked healthy establishments, and 94.4% (*n* = 17) had unhealthy ones; only one school (5.6%) had no establishments in this buffer. All schools had at least one healthy and one unhealthy establishment within 800 m. No significant differences were observed in the presence of at least one healthy or unhealthy establishment by school type or social vulnerability.

In the 250 m buffer, the density of unhealthy establishments was higher in low-vulnerability areas (median = 31.9) than in medium/high-vulnerability areas (median = 5.5; *p* = 0.03). In the 800 m buffer, low-vulnerability areas also had higher densities of healthy establishments (median = 19.9 vs. 7.9; *p* = 0.01), unhealthy establishments (median = 23.6 vs. 8.5; *p* < 0.01), and total density (median = 40.4 vs. 17.3; *p* < 0.01). Public schools presented higher total density in the 800 m buffer than private schools (median = 41.0 vs. 21.0; *p* < 0.01), with a significant difference for unhealthy establishments (median = 24.6 vs. 9.6; *p* < 0.01) (Table 2).

Figure 1 presents the spatial distribution of the establishments, highlighting differences due to the type of school. Healthy (green dots) and unhealthy (red triangles) food outlets are shown within 250 m, 400 m, and 800 m buffers around public and private schools in selected administrative regions of the Federal District, Brazil. Private schools (orange buffers) tend to be located in areas with a greater concentration of food establishments overall, while public schools (blue buffers) are more often situated in areas with fewer surrounding outlets. In some regions, buffer overlap is evident, particularly in central areas where schools are close to each other, leading to some establishments being repeated in the counts. The spatial pattern aligns with the quantitative results, showing a higher proportion of food outlets around private schools in the 800 m buffer, with differences less marked at shorter distances.

### 3.2. Internal and Surrounding Food Environment According to School Staff Reports

The characteristics of both the internal and vicinity food environment, as reported by the school principal or pedagogical coordinator, were similar across school types (Table 3) and levels of social vulnerability (Table 4).

All schools with canteens were located in areas of low social vulnerability. Canteens and/or restaurants were reported in 55.6% (95% CI: 21–85.4) of private schools, and 50% (95% CI: 19.3–80.7) of those were located in low-vulnerability areas. In one half, the acquisition of food was performed directly by the students and, in the other half, through a payment by the parents beforehand.

Half the schools reported conducting Food and Nutrition Education (FNE) activities, with no statistically significant difference by school type and social vulnerability. Most schools provided 20 to 30 min for meals. Nutritionists were present in 44.4% (95% CI: 12.5–79) of private schools; in public schools, nutritionists were from the National School Feeding Program (PNAE).

Seven schools (41.2%; 95% CI: 19.7–66.5) reported the presence of informal food outlets in the vicinity (e.g., street vendor or cart)—three private (37.5%; 95% CI: 9.6–77.1) and four public (44.4%; 95% CI: 14.5–79)—with no statistically significant difference (Table 3).

Medians of 6.0 UPF and 2.5 MPF items were reported in canteens. In the vicinity, medians of 4.0 UPF and 1.0 MPF items were observed. The proportion of MPF items was 83.3% in the canteens and 25% in the vicinity; the proportion of UPF items was 60% and 40%, respectively. The commercialization of UPF was 170% higher than the total number of MPF items in the canteens and 200% higher in the vicinity. The HI of private canteens presented a median of 38.5, while the HI of the surroundings had a median of 50.0. There was no significant difference according to the type of school (Table 3) and social vulnerability (Table 4).

In the private school canteens, the most frequent foods were savory pastries (with meat, chicken, and/or cheese) and fresh fruits and/or natural fruit juice and/or pulp (87.5%), followed by sugar-sweetened beverages and popsicles and/or ice cream (75%), tapioca and/or couscous, savory pastries (with ham, pepperoni, hamburger meat, and/or sausage), chips and/or microwave popcorn and/or industrialized sweet popcorn and pizza (62.5%). Among the least available foods were energy drinks (present in only 12.5% of schools), soft drinks (25%), and hamburgers (25%).

In the vicinity, the most cited items were chips and/or microwave popcorn and/or industrialized sweet popcorn (57.1%), sugar-sweetened beverages (42.9%), churros (42.9%), popsicles and/or ice cream (42.9%), and soft drinks (42.9%). Savory pastries with meat and/or chicken and savory pastries with ham, pepperoni, hamburger meat, and/or sausage (28.6%), hamburgers (28.6%), boxed lunch (28.6%), and açaí (28.6%) were also reported. Fresh fruits, gelatin, noodles, cupcakes, and energy drinks were not reported in the vicinity by any school.

There was no statistically significant difference between food availability in the vicinity and social vulnerability or type of school and between food in the canteen and social vulnerability.

## 4. Discussion

This study is among the first to map the food environment around schools in the Federal District of Brazil. The results revealed that there were many (40.2%) establishments in which students were exposed to predominantly ultra-processed products, and 49.4% were classified as unhealthy.

The majority (83.3%) of schools had at least one point of sale within a 250 m radius, with 77.8% having at least one unhealthy point and 38.9% having no healthy options. The findings are consistent with a study conducted in Viçosa, Minas Gerais, Brazil, which also found a high concentration of food outlets in areas near schools [43]. In Belo Horizonte, Minas Gerais, a survey revealed that 97.4% of the schools evaluated had at least one commercial establishment within 250 m [25].

The analysis also showed that social vulnerability was associated with the availability of establishments. Schools in less vulnerable areas had more establishments, healthy or unhealthy, corroborating studies that associate higher income with a higher concentration of commercial establishments [25,43,44]. Private schools had a higher density of points of sale, especially of unhealthy food, within a radius of 800 m. Similarly, Novaes et al. [43] reported that private schools in higher-income areas had more establishments selling food in the 400 m and 800 m buffers. Data from the Cardiovascular Study in Adolescents (ERICA—Estudo Cardiovascular em Adolescentes) corroborate this pattern, showing that private schools have more obesogenic food environments than public schools [45]. The Study on Food Commercialization in Brazilian Schools (Caeb—Estudo de Comercialização de Alimentos em Escolas Brasileiras) reinforced this scenario, pointing out that the average number of UPFs sold in private schools was 50% higher than MPF.

However, international evidence contrasts with the findings of the present study. Research conducted in cities such as Madrid, New York, Mexico City, and Santiago has shown that schools located in lower-income areas tend to be more exposed to unhealthy food retailers [46,47,48,49]. Although 90% of schools in Barcelona had at least two unhealthy food outlets nearby, those located in higher-income neighborhoods had a significantly greater availability and affordability of healthy foods [50]. These findings highlight persistent socioeconomic disparities in school food environments across diverse global settings. Addressing such inequalities requires comprehensive and context-sensitive public policies that regulate the food environment both inside and around schools.

In half of the private schools analyzed in this study, the students bought food from the canteen or restaurant themselves. As a phase of greater autonomy and the beginning of financial freedom [51], adolescence can be a period of greater vulnerability to unhealthy food choices due to a lack of nutritional knowledge, culinary skills, or financial autonomy [52,53]. Thus, schools play an essential role in promoting healthy habits, both by providing adequate food and through FNE [52,53].

Only half of the analyzed institutions carry out FNE, even though it is fundamental for the formation of healthy eating habits [52]. The mandatory FNE in elementary and high school curricula, according to Law No. 13,666 of 16 May 2018 [54], represents an advance. However, to promote effective and sustainable changes, this policy must be combined with broader actions, considering the environmental and structural determinants that influence the diet of young people [51].

The World Health Organization (WHO) emphasizes the importance of mandating food and health education in the core school curriculum as a strategy to strengthen nutrition literacy and develop healthy eating skills among students, parents, and caregivers. However, to promote effective and lasting changes, educational efforts must be accompanied by structural interventions in the school food environment. Thus, the WHO also recommends that governments establish clear nutritional standards for school meals and for foods sold on school premises, ensuring alignment with healthy eating guidelines. It further advises restricting the sale and marketing of unhealthy products in schools and creating buffer zones around them to limit children’s exposure to obesogenic environments. These measures highlight the essential role of the education sector in addressing childhood obesity and fostering healthy habits from an early age [55].

Australia, Bulgaria, Chile, Canada, Costa Rica, South Korea, Ecuador, Estonia, France, Hungary, Mexico, Poland, and the United Kingdom have adopted regulations about the sale and advertising of food in school canteens. These policies aim to restrict the availability and marketing of unhealthy foods and promote healthier dietary practices among students. The diversity of national approaches highlights a global recognition of the school environment as a key setting for nutrition-related interventions and underscores the importance of regulatory frameworks to protect children’s health [56].

In the normative scope, District Decree No. 36,900/2015 [57] seeks to promote adequate and healthy food in DF schools by restricting the sale of unhealthy food in canteens and within a radius of 50 m around educational institutions. Furthermore, Presidential Decree No. 11,821.0/2023 [58] established national guidelines for the promotion of adequate and healthy food in the school environment, determining that states, municipalities, and the Federal District should implement their own regulations and actions, in line with the Dietary Guidelines for the Brazilian Population. This highlights the need to update district legislation to incorporate national guidelines and ensure effective implementation [59].

However, all private schools analyzed failed to comply with the regulations, offering at least one item on the list of unhealthy foods, highlighting potential weaknesses in the current regulation and the need for its review and enhanced inspection. In Curitiba, a similar result showed that two-thirds of the canteens did not comply with the Healthy Canteen Law of 2005 [60] and had a high prevalence of non-compliant foods in canteens, such as industrialized snacks, chocolates, candies, filled cookies, and artificial juices [61].

The implementation of school food regulations faces significant challenges, particularly in private institutions. These schools often show greater resistance to compliance, partly due to their limited integration into public food and nutrition policies, such as the National School Feeding Program (PNAE), which facilitates adherence in public schools [59,62]. The effectiveness of regulatory measures is further weakened by the lack of educational campaigns, limited intersectoral coordination, political and economic pressures from the private sector, and insufficient monitoring systems [62]. In the Federal District, although Decree No. 36,900 of 23 November 2015 [57] assigns the Health Department the responsibility to inspect school canteens [63]; however, no systematic mechanisms for monitoring and enforcement have been effectively implemented. Although these barriers were not directly assessed in the present study, they are well documented in the literature and provide important context for understanding the limited enforcement and compliance observed in some schools [59,62,64].

In this study, the private school canteens mainly offered MPF, such as savory pastries with meat, chicken, and/or cheese and fresh fruits, and natural fruit juice or pulp, and UPF, such as sweetened drinks, popsicles and ice cream, savory pastries filled with sausages, chips, microwave popcorn or industrialized sweet popcorn, and pizza. These patterns reflect food consumption outside the home in Brazil, with a predominance of savory snacks (fried and baked), followed by sweetened beverages, ice cream, and chips [33].

The HI was 38.5 in the canteens and 50 in the surroundings, revealing that changes are still needed to make these environments healthier. Caeb data showed a mean HI in Brazilian canteens of 56.6 and in the DF of 60.69 (59.44–61.95). The mean proportions were 28.9% for MPF (33.04 in DF) and 23.3% for UPF (19.28 in DF) [65]. It is important to note that the methodologies differ between the two studies, including the list of subgroups considered, the cut-off criteria, and the data collection period, which may influence the comparability of the results. In the present study, the results differed: PROP-MPF was 83.3% and PROP-UPF was 60%. On average, the total number of UPF items offered was 2.7 times (170%) higher than that of MPF in the canteens and 3.0 times (200%) higher in the surrounding area. Despite their differences, both studies indicate a greater presence of UPF than MPF in canteens.

A previous study carried out in the Federal District already indicated that most school canteens were not spaces that facilitated healthy eating [37]. In these places, unhealthy foods are widely visible and accessible, thus playing an important role in shaping the food environment and establishing unhealthy eating habits [66]. Research by Porto et al. [37] was carried out before the implementation of district legislation aimed at promoting healthy eating in schools; nevertheless, our study indicates that challenges persist.

Evidence from an analysis of policies implemented in Brazilian capitals indicated that state and local regulations can effectively reduce the availability of unhealthy foods, especially in more developed regions and public schools, which benefit from the PNAE. However, the effectiveness of these laws depends on the supervision and implementation of mechanisms to ensure compliance, in addition to offering healthier alternatives to students [67]. Therefore, improving school feeding must be a continuous and joint effort, involving multiple actors [37].

Machado and Höfelmann [60] also highlight the need to broaden the view of the school food environment inside and outside the school because as regulations on canteens become stricter, the marketing of unhealthy foods may increase around schools. Thus, policies must be combined with broader territorial strategies.

In the surroundings, the most mentioned items were chips and/or microwave popcorn and/or industrialized sweet popcorn, sugar-sweetened beverages, churros, popsicles and/or ice cream, and soft drinks. Although no difference was found regarding type of school and SVI, PeNSE data showed that, at alternative points of sale in public schools, the most frequently marketed items were soda, packaged snacks, and fried snacks, while in private schools, they were candies, confectionery, sweets, and others, sweetened beverages, and packaged snacks [68]. This disparity reinforces social inequalities and contributes to the maintenance of unhealthy eating habits among the most vulnerable students, perpetuating an obesogenic food environment.

## 5. Conclusions

The mapping and characterization of the food environment around and within schools in the Federal District found challenges and inequalities in the supply of healthy food. Adolescents are potentially exposed to a predominantly unhealthy food environment, with differences observed according to income and school type. Schools in higher-income areas had more food establishments in their surroundings. Private schools had more points of sale and offered ultra-processed foods in the canteen, violating existing regulations. The scarcity of FNE initiatives also contributes to this scenario.

These findings highlight the need for strategies that promote healthier school food environments, including updating and enforcing current regulations, expanding the supply of suitable food in canteens and the school environment, and strengthening educational actions aimed at healthy eating to encourage adolescents’ autonomy to make better choices. This study also contributed to understanding the food environment in the Federal District and can support more effective public policies adapted to the local reality.

### Study Limitations and Future Research

Limitations of this study include the absence of on-site data on informal street vendors, whose presence was reported by school representatives. Due to the informal and dynamic nature of this type of commerce, students’ real exposure to these food sources was probably underestimated. The use of administrative data (RAIS) and virtual tools (Google Street View) may not fully reflect current operations or food offerings at the time of student access. However, a detailed virtual audit was conducted to verify the existence, location, and facade of establishments, including CNAE corrections and the inclusion of unlisted eligible outlets.

The use of Euclidean (straight-line) distances to define 250 m, 400 m, and 800 m buffers may not correspond to actual routes taken by students, especially in urban areas with physical barriers such as highways, a lack of sidewalks, or unsafe crossings. Although this technique is common in geographic studies for its simplicity and comparability, future studies could apply network-based distances to improve precision.

This study also did not assess students’ actual food consumption or preferences, which limits the interpretation of how the food environment affects behavior. Additionally, data on the in-school environment were based on reports from school representatives, which may be subject to social desirability bias. Investigating purchasing motivations and decision-making would provide important context and is recommended for future research. The cross-sectional design also limits the ability to account for temporal variations, such as seasonal vendors or changes in food offerings throughout the year. Data collection at multiple time points, particularly before and after the implementation of new regulations, is suggested.

Although two private schools declined participation, refusals did not appear to be associated with identifiable institutional characteristics. As with most probability samples, it was not possible to directly compare respondents and non-respondents, and we assumed no systematic differences between these groups. While methodologically acceptable, this assumption introduces a potential, albeit limited, source of selection bias.

Finally, while this study was conducted in a region with specific socioeconomic and institutional characteristics, its findings may reflect patterns common to other Brazilian urban centers. Broader, multicenter, and longitudinal studies are encouraged to confirm and expand these findings in diverse contexts and to strengthen territorial characterization and compliance monitoring.

On the other hand, the comprehensive assessment of the school food environment in a little-explored Brazilian region, the methodological rigor in the classification and validation of establishments, and the comparative analyses according to school type and social vulnerability are strengths of this study.

## Figures and Tables

**Figure 1 ijerph-22-01331-f001:**
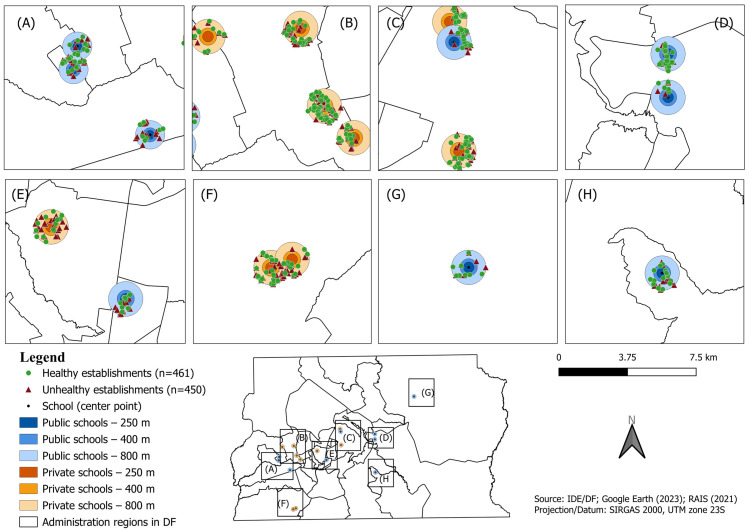
Spatial distribution of healthy (green dots) and unhealthy (red triangles) food establishments within 250 m, 400 m, and 800 m buffers around public (blue) and private (orange) schools in selected administrative regions of Federal District, Brazil (2021).

**Table 1 ijerph-22-01331-t001:** Proposed classification of foods sold in the school canteen and its surroundings.

Local	Fresh, Minimally Processed, or Processed Foods (MPFs) and Culinary Preparations Based on These Foods	Ultra-Processed Foods (UPFs)
School canteens	(1) Fresh fruits, natural fruit juice, and pulp juice (2) Tapioca and couscous (3) Savory pastries (with meat, chicken, and/or cheese)	(1) Sugar-sweetened beverages (box juice, powdered juice, chocolate drinks, teas, and sports drinks) and energy drinks (2) Filled and unfilled cookies (3) Sweets: industrialized cakes, chocolates, candy, sweets, lollipops, and gum; Brigadeiro; and cupcakes (4) Hamburger (5) Popsicles and ice cream and açaí (6) Pizza (7) Soft drinks (8) Chips, microwave popcorn, and industrialized sweet popcorn (9) Savory pastries (with ham, pepperoni, hamburger meat, and/or sausage) (10) Gelatin
Surroundings of the schools	(1) Tapioca and couscous (2) Savory pastries (with meat, chicken, and/or cheese) (3) Boxed lunch with a full meal (4) Popcorn	(1) Sugar-sweetened beverages (box juice, powdered juice, chocolate drinks, teas, sports drinks) and soft drinks (2) Filled and unfilled cookies (3) Sweets: industrialized cakes, chocolates, candy, sweets, lollipops, and gum and Brigadeiro (4) Hamburger (5) Popsicles and ice cream and açaí (6) Pizza (7) Soft drinks (8) Chips, microwave popcorn, and industrialized sweet popcorn (9) Savory pastries (with ham, pepperoni, hamburger meat, and/or sausage) (10) Churros

**Table 2 ijerph-22-01331-t002:** Density of food sales establishments in the school environment by social vulnerability and type of school, within 250 m, 400 m, and 800 m buffers. Federal District, 2022–2023.

Buffer	Classification of the Establishments	Median Density (Estab./10 K Inhab.) (IQR)		*p* ^1^	Median Density (Estab./10 K Inhab.) (IQR)		*p* ^2^
		Low	Medium/High		Private	Public	
250 m	Healthy	21.6	0	0.17	24.6	0	0.22
		(20.4)	(19.0)		(20.4)	(18.5)	
	Unhealthy	31.9	5.5	0.03 *	39.2	9.7	0.06
		(42.3)	(14.5)		(40.9)	(8.1)	
	Total	50.8	14.5	0.05	52.3	16.9	0.10
		(61.3)	(26.2)		(61.3)	(20.0)	
400 m	Healthy	17.1	7.6	0.08	21.2	9.4	0.06
		(37.4)	(14.0)		(36.8)	(7.2)	
	Unhealthy	22	7.3	0.09	23.5	5.2	0.08
		(39.1)	(8.3)		(34.8)	(6.8)	
	Total	41.6	13.8	0.05	46.6	15.9	0.03 *
		(84.3)	(17.7)		(70.5)	(12.4)	
800 m	Healthy	19.9	7.9	0.01 *	19.5	9.6	0.07
		(19.1)	(9.6)		(19.1)	(11.6)	
	Unhealthy	23.6	8.5	<0.01 *	24.6	9.6	<0.01 *
		(9.7)	(6.02)		(7.2)	(3.5)	
	Total	40.4	17.3	<0.01 *	41.0	21.0	<0.01 *
		(21.3)	(14.6)		(21.3)	(15.1)	

Values in parentheses indicate the interquartile range (IQR); Total refers to the sum of the densities of healthy and unhealthy food retail outlets; the *p* ^1^ value compares the density distributions of the establishments and the levels of social vulnerability (Mann–Whitney test); the *p* ^2^ value compares the density distributions of the establishments and the type of school (Mann–Whitney test). * *p* < 0.05.

**Table 3 ijerph-22-01331-t003:** Characteristics of the food environment of the participating schools, according to the type of school. The Federal District, 2022–2023.

			Type of School				*p* ^1^
Schools’ Characteristics	Total (%)	95% CI: Lower–Upper	Private (*n* = 9) (%)	95% CI: Lower–Upper	Public (*n* = 9) (%)	95% CI: Lower–Upper	
**Presence of canteen and/or restaurant**							
Canteen and/or restaurant	27.8	1.1–54.0	55.6	21–85.4	0	-	-
Canteen only	16.7	5.0–43.1	33.3	8.9–71.8	0	-	
None	55.6	31.5–77.3	11.1	1.1–59	100	-	
**Presence of microwaves for students**	55.6	31.5–77.3	77.8	35.5–95.7	33.3	8.9–71.8	0.15
**Type of food supply for students in institutions with canteens**					**		
Students shop at the canteen and/or restaurant	50	15.8–84.2	50	15.8–84.2	N/A	-	-
Parents prepay	50	15.8–84.2	50	15.8–84.2	N/A	-	
**Authorization to take snacks from home or purchase outside of school**	83.3	56.8–94.5	88.9	40.9–99	77.8	35.5–95.7	1.00
**Performs FNE activities**	50	27–72.9	55.6	21–85.4	44.4	14.5–79	1.00
**Presence of nutritionist**							
Yes	72.2	46.1–88.7	44.4	12.5–79	100	-	-
No	22.2	7.9–48.6	44.4	12.5–79	0	-	
Do not know	5.6	0.6–34	11.2	1.1–59	0	-	
**Where meals are eaten**							
Cafeteria	55.6	31.5–77.3	4	14.5–79	6	28.1–91.1	0.15
Canteen area	22.2	7.9–48.6	4	14.5–79	0		
Classroom	22.2	7.9–48.6	1	1.1–59.1	3	8.9–71.8	
**Time available for meals**							
10–15 min	22.2	7.9–48.6	22.2	4.3–64.4	22.2	4.3–64.4	1.00
20–30 min	55.6	31.5–77.3	55.6	21–85.4	55.6	21–85.4	
>30 min	22.2	7.9–48.6	22.2	4.3–64.4	22.2	4.3–64.4	
**Presence of alternative points of sale for food and/or beverages at door or nearby**							
Yes	41.2	19.7–66.5	37.5	9.6–77.1	44.4	14.5–79	1.00
No	47.1	21.1–71.3	50	15.8–84.2	44.4	14.5–79	
Do not know	11.8	2.6–39.7	12.5	1.1–64.1	11.1	1.1–59	
**Indicators**	**Total**		**Type of School**				***p* ^2^**
			**Private**		**Public**		
HI of food offered in school canteen	38.5		38.5		N/A		-
	(30.8–84.6)		(30.8–84.6)				
UPF/MPF ratio canteen	2.7		2.7		N/A		-
	(0.5–6.0)		(0.5–6.0)				
HI of food offered around schools	50		50		46.4		0.86
	(28.6–64.3)		(50–57.1)		(28.6–64.3)		
UPF/MPF ratio surroundings	3		3		5.25		0.25
	(2.5–4)		(2.5–4)		(3.5–7.0)		

95% *CI* = confidence interval of proportion; FNE = Food and Nutrition Education; HI = Healthiness Index; N/A = not applicable; ^1^ Fisher’s exact test; ^2^ Mann–Whitney U test; School characteristics are presented as proportions. Indicators are presented as median (minimum–maximum). ** In public schools, meals are provided free of charge through the National School Feeding Program (PNAE). For the same sample, different stratifications are presented for readability. The variables and definitions are identical to those in Table 4.

**Table 4 ijerph-22-01331-t004:** Characteristics of the food environment of the participating schools, according to social vulnerability. The Federal District, 2022–2023.

			Social Vulnerability				*p* ^1^
Schools’ Characteristics	Total (%)	95% CI: Lower–Upper	Low (*n* = 10) (%)	95% CI: Lower–Upper	Medium/High (*n* = 8) (%)	95% CI: Lower–Upper	
**Presence of canteen and/or restaurant**							
Canteen and/or restaurant	27.8	1.1–54.0	50	19.3–80.7	0	-	-
Canteen only	16.7	5.0–43.1	30	8.2–67.1	0	-	
None	55.6	31.5–77.3	20	4–59.9	100	-	
**Presence of microwaves for students**	55.6	31.5–77.3	80	40–96	25	4.6–69.7	0.05
**Type of food supply for students in institutions with canteens**							
Students shop at the canteen and/or restaurant	50	15.8–84.2	50	15.8–84.2	0		-
Parents prepay	50	15.8–84.2	50	15.8–84.2	0		
**Authorization to take snacks from home or purchase outside of school**	83.3	56.8–94.5	90	45.3–99	75	30.3–95.4	0.56
**Performs FNE activities**	50	27–72.9	50	15.8–84.2	50	15.8–84.2	1.00
**Presence of nutritionist**							
Yes	72.2	46.1–88.7	60	25.8–86.6	87.5	35.8–98.9	0.23
No	22.2	7.9–48.6	30	8.2–67.1	12.5	1.1–64.1	
Do not know	5.6	0.6–34	10	1–54.7	0	-	
**Where meals are eaten**							
Cafeteria	55.6	31.5–77.3	60	25.8–86.6	50	15.8–84.2	
Canteen area	22.2	7.9–48.6	40	13.4–74.2	0	-	0.01 *
Classroom	22.2	7.9–48.6	0	-	50	15.8–84.2	
**Time available for meals**							
10–15 min	22.2	7.9–48.6	30	8.2–67.2	12.5	1.1–64.1	0.55
20–30 min	55.6	31.5–77.3	40	13.4–74.2	62.5	30.3–95.4	
>30 min	22.2	7.9–48.6	30	8.2–67.2	12.5	1.1–64.1	
**Presence of alternative points of sale for food and/or beverages at door or nearby**							
Yes	41.2	19.7–66.5	44.4	14.5–79	37.5	9.6–77.1	1.00
No	47.1	21.1–71.3	44.4	14.5–79	50	15.8–84.2	
Do not know	11.8	2.6–39.7	11.1	1.1–59	12.5	1.1–64.1	
**Indicators**	**Total**		**Social Vulnerability**				***p* ^2^**
			**Low**		**Medium/High**		
HI of food offered in school canteen	38.5		38.5		N/A		-
	(30.8–84.6)		(30.8–84.6)				
UPF/MPF ratio canteen	2.7		2.7		N/A		-
	(0.5–6.0)		(0.5–6.0)				
HI of food offered around schools	50		53.6		35.7		0.20
	(28.6–64.3)		(50–64.3)		(28.6–57.1)		
UPF/MPF ratio surroundings	3		5.25		3		0.25
	(2.5–4)		(3.5–7)		(2.5–4)		

95% *CI* = confidence interval of proportion; FNE = Food and Nutrition Education; HI = Healthiness Index; N/A = not applicable; ^1^ Fisher’s exact test; ^2^ Mann–Whitney U test; School characteristics are presented as proportions. Indicators are presented as median (minimum–maximum). * *p* < 0.05.

## Data Availability

The data presented in this study are not publicly available but are available from the corresponding author upon reasonable request.

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
