# Peer review of "What Is There to Buy? An Analysis of the Food Environment in Public and Private Schools in the Federal District"

_ijerph, 2025, doi:10.3390/ijerph22091331_

Round 1
Reviewer 1 Report (Previous Reviewer 3)
Comments and Suggestions for Authors
The authors made appropriate changes, so the paper can be published in a current form.
Comments on the Quality of English LanguageMinor editing is needed
Author Response
- Summary
We sincerely thank the reviewer for taking the time to evaluate our manuscript and for the positive assessment indicating that it can be published in its current form. We have addressed the comment on the quality of English by performing a final careful proofreading to enhance clarity, grammar, and consistency. The revised manuscript incorporates these minor edits, which are marked in track changes in the re-submitted version.
- Questions for General Evaluation
|
Reviewer’s Evaluation Question |
Reviewer’s Evaluation |
Response and Revisions |
|
Does the introduction provide sufficient background and include all relevant references? |
Can be improved |
The introduction was re-read carefully; no missing references were identified, but minor edits were made to improve clarity and sentence flow. |
|
Is the research design appropriate? |
Can be improved |
Research design was maintained, but descriptions were lightly revised for clarity. |
|
Are the methods adequately described? |
Can be improved |
Minor adjustments in wording were made to improve clarity in the methods section. |
|
Are the results clearly presented? |
Can be improved |
Minor editing of wording and sentence structure for clarity. |
|
Are the conclusions supported by the results? |
Can be improved |
No changes in content; minor language refinements applied. |
|
Are all figures and tables clear and well-presented? |
Can be improved |
Legends were checked for clarity and consistency; minor adjustments made where necessary. |
- Point-by-point response to Comments and Suggestions for Authors
Comment 1:
The authors made appropriate changes, so the paper can be published in a current form.
Response 1:
We thank the reviewer for this positive evaluation and for recognizing the improvements made in the previous revision.
- Response to Comments on the Quality of English Language
Point 1:
Minor editing is needed.
Response 1:
The manuscript underwent a final proofreading by a professional translator and native English speaker, as certified in the attached English Certification (ID: 92104.297-276, August 14, 2025). This review focused on improving clarity, grammar, and consistency throughout the text.
All these minor revisions are highlighted in track changes in the re-submitted manuscript.
- Additional clarifications
No further clarifications are required at this time.
Reviewer 2 Report (Previous Reviewer 4)
Comments and Suggestions for Authors
The authors have adequately addressed all the requested modifications. The revised manuscript meets the expected standards, and I have no further comments.
Author Response
Reviewer 2
We sincerely thank the reviewer for the positive feedback and for acknowledging that the revised manuscript meets the expected standards. We appreciate the recognition of our efforts to address the previous comments and improve the quality of the work. No further changes were made in response to this round of review.
The manuscript underwent a final proofreading by a professional translator and native English speaker, as certified in the attached English Certification (ID: 92104.297-276, August 14, 2025). This review focused on improving clarity, grammar, and consistency throughout the text.
All these minor revisions are highlighted in track changes in the re-submitted manuscript.
Reviewer 3 Report (New Reviewer)
Comments and Suggestions for Authors
General comment
The study provided a comprehensive report on the hygiene and safety of food retail outlets (FRO) situated within the radius of private and public schools in the Federal District of Brazil. The authors have made considerable effort in highlighting the healthy state of these FROs, the commonly available food, and the SVI.
Having read the content of the article, I want to suggest that a public health analyst is also invited to review and provide more clarity. However, I have added some comments that may improve the article.
Specific comments
Title
What is there to buy? Analysis of the food retail environment around (or servicing) public and private schools in the Federal District of Brazil
Abstract
A few details need to be included in the abstract to provide more clarity about the school types, the radius of study, and the SVI, as spelt out in the objectives
- Which school type had the most abundant food retail outlets (FRO)? Kindly include the percentage by radius
- Include the most abundant ultra-processed product
- It’s best for the SVI for the school types to be provided in the abstract
Keywords
- Some suggested keywords are attached
Food hygiene and safety, food monitoring, Food contamination
Introduction, materials and methods, and discussion: The authors are advised to make good efforts by improving the grammar in these sections for clarity.
Results: The authors should consider including the confidence interval of proportion for all percentages in the results
Comments on the Quality of English LanguageSome sections of the article were poorly written with several grammatical errors. The authors are advised to read the article carefully, correct redundant words, rephrase, and improve the grammatical expression in the manuscript.
Author Response
Response to Reviewer 3 Comments
- Summary
We sincerely thank the reviewer for the careful evaluation of our manuscript and for the constructive comments and suggestions, which have helped to further improve the clarity and quality of the paper. We have revised the manuscript accordingly, with all changes highlighted in track changes in the re-submitted version.
- Questions for General Evaluation
|
Reviewer’s Evaluation Question |
Reviewer’s Evaluation |
Response and Revisions |
|
Does the introduction provide sufficient background and include all relevant references? |
Must be improved |
The Introduction section was carefully revised. |
|
Is the research design appropriate? |
Yes |
No changes in design; methods clarified where necessary. |
|
Are the methods adequately described? |
Yes |
Minor rewording for clarity. |
|
Are the results clearly presented? |
Yes |
Minor edits applied; confidence intervals for proportions were added where applicable. |
|
Are the conclusions supported by the results? |
Can be improved |
Conclusion strengthened by linking findings more directly to results and policy implications. |
|
Are all figures and tables clear and well-presented? |
Yes |
Figures and tables checked for clarity; captions revised for precision. |
- Point-by-point response to Comments and Suggestions for Authors
General comment:
The study provided a comprehensive report… the authors have made considerable effort… I want to suggest that a public health analyst is also invited to review…
Response:
We appreciate the positive feedback on the scope and comprehensiveness of the study. The manuscript has undergone additional review by our research team, which includes public health specialists, to ensure the accuracy and clarity of the public health concepts presented.
Specific comments:
Title:
Suggested modification to “What is there to buy? Analysis of the food retail environment around (or servicing) public and private schools in the Federal District of Brazil”
Response:
We appreciate the suggestion. However, after careful consideration, we have decided to maintain the original title. The proposed version, while clear, would not fully encompass the study’s methodology, which includes the assessment of both the external food environment (around schools) and the internal food environment (within schools). The original title was chosen to preserve brevity, maintain alignment with the journal’s style, and ensure clarity for an international audience.
Abstract:
Requests: include details on school types, radius, and SVI; indicate which school type had the most abundant FROs and percentages by radius; include the most abundant ultra-processed product; provide SVI by school type.
Response:
We have revised the abstract to:
- Clarify that the study analyzed 18 public and private schools, with food retail outlets assessed within 250 m, 400 m, and 800 m buffers. (Lines 20 - 21)
- Indicate that private schools had the highest proportion of surrounding FROs. (Lines 26 - 27)
- Include the most frequent ultra-processed item in canteens (sugar-sweetened beverages). (Lines 30-31)
- Present the distribution of school types by SVI. (Line 26)
Keywords:
Suggested keywords: Food hygiene and safety, food monitoring, Food contamination
Response:
We appreciate the suggestions; however, the scope of our study did not include microbiological analysis or direct food safety assessments. Therefore, these terms may not accurately reflect the study’s focus. We maintained the original keywords aligned with the research scope but ensured they reflect key study elements.
Introduction, materials and methods, and discussion – grammar improvement:
Response:
We have conducted a comprehensive proofreading of these sections, correcting grammatical errors, improving sentence flow, and removing redundant words to enhance clarity.
Results – add confidence intervals of proportion:
Response:
We have included the 95% confidence intervals for proportions in all relevant results, as per the suggestion.
- Response to Comments on the Quality of English Language
Point 1:
Some sections were poorly written with several grammatical errors…
Response 1:
The manuscript underwent a final proofreading by a professional translator and native English speaker, as certified in the attached English Certification (ID: 92104.297-276, August 14, 2025). This review focused on improving clarity, grammar, and consistency throughout the text.
Changes include:
- Rewriting sentences for improved readability.
- Correcting redundant words.
- Standardizing technical terms.
- Ensuring consistent formatting and terminology throughout.
All these minor revisions are highlighted in track changes in the re-submitted manuscript.
- Additional clarifications
No further clarifications are required.
Reviewer 4 Report (New Reviewer)
Comments and Suggestions for Authors
Summary of the paper
The Authors evaluate the food environment of public and private schools in Brazil, focusing on how the food environment is configured inside and around the schools in the Federal District (DF), and how configuration is associated with social vulnerability. In the Introduction, the Authors provide the importance of the topic, and justification of study. Also, they state the research questions and research gap supported by concise literature review, as well as the main contribution of the study. In the Methods section, they detail the sample size and sampling technique used, study area, and methods of analyses. In the Results section, they present results on the school surroundings characterization (Table 1 & Figure 1), and internal and surrounding food environment based on data obtained from school heads (Table 2). In the Discussion section, the Authors provide discussion on results, limitations of study and proposed topics for future research. In the conclusion section, they provide a summary of the findings and limited policy advice.
Overall evaluation and key concerns
The paper is well-structured. It provides interesting reading on a key topic in the agri-food system-school food environment and food availability. The main contribution lies in the dataset and in a comprehensive analysis, allowing Authors to conclude that there are challenges and inequalities in the supply of healthy food around and within schools in the Federal District of Brazil. Given that school food environment and food availability is a subject of continuous research, and policy discussions, carrying out this analysis is apt.
However, some important issues must be addressed before the manuscript can be considered for publication. I summarize my concerns in the following points:
Introduction.
- Term definition missing: Authors might want to provide a clear definition of social vulnerability. While the meaning of social vulnerability in the context of food environment can be deduced from the text in the introduction, not clearly stating it may make reading and comprehension difficult for the broad international audience of the journal. A sentence defining it, after the first mention of social vulnerability in the text (Line 74) would be enough.
- Authors might want to provide the structure of the paper at the end of this section to give readers a fair idea of the content of subsequent sections. A couple of sentences describing the organisation of the paper would be enough.
- Line 50-51: Please provide reference to support the statement.
Methods.
- Authors write (Line 90-92), “The Human Development Index (HDI) is considered very high at 0.824, [18], and the Social Vulnerability Index (SVI) is classified as average at 0.33 [14]. Since your analysis considered the SVI, you might want to provide a description/definition of the Index and what an average classification means.
- Sample size: The Authors write “Although the small sample size may limit the statistical precision of the estimates, the probabilistic design ensures that the sample is representative of the target population.” There is no mention of the exact sample size used prior to this sentence. In fact, Authors only stated the study population. I suggest clearly stating the sample size before the sentence.
- Equations: Please number all equations. This will provide clarity, making reading interesting.
- Line 95-97: The sentence is not clear. Please double check.
- Line 108-119: Please provide reference/s to support sentences.
- Line 118-121: Please can you provide a justification or the importance of strictly following the steps for identification and classification of private food retail establishment? For example, the steps limited the risk of missing out on retail establishments etc. Alternatively, you can reference previous authors who followed such approach.
- Line 123: Please take out “The” and reframe the sentence. This is because, there was no mention of “selected codes” prior to the sentence.
- Line 141 and 144: The use of “according” appears twice here, making the entire sentence unclear. Please double check and make corrections.
- Box 1: I suggest using a Table to clearly present items.
Results.
- Table 1: Line 295 to 303 should be right under Table 1. At the moment, the sentences appear to be isolated from the table.
- Inconsistency: Similar to Table 1, the title of Figure 1 (310 to 312) should be at the top of the figure.
- Figure 1: Authors provided figures without providing any text to explain or clearly communicate the results. Please address this.
- Table 2: See comment on Table 1.
- Authors write (line 344-345) “PROP-MPF was 83.3% in the can- 344 teens and 25% in the surroundings; PROP-UPF was 60% and 40%, respectively (data not shown).” Can you clarify what you mean by “data not shown”.
- Line 367: See comment on line 344-345.
Discussion.
- Duplication of text: Line 448 to 460 and Line 461 to 473 have the same text. Please delete one and double check the intext citations you used.
- Line 495-497: Authors write “The analysis of policies implemented in Brazilian capitals showed that state and local regulations can effectively reduce the availability of unhealthy foods …”. To the best of my understanding, you did not analyse policies. You might want to clarify this.
- I suggest moving text on limitations of study as well as proposed future research topics to the conclusion section, and solely discussing results in the context of prior studies in this section.
- Proof reading required.
Conclusion.
- See comment under discussion (point 4).
Minor Comment: Proofreading required.
Comments on the Quality of English Language
Proofreading is required to correct grammatical errors.
Author Response
Response to Reviewer 4
Comments
We sincerely thank the reviewer for the positive and constructive feedback on our manuscript. We appreciate the recognition of the study’s relevance, contribution, and structure, and we have carefully addressed each of the points raised. All modifications are highlighted in track changes in the revised manuscript.
Questions for General Evaluation
|
Reviewer’s Evaluation Question |
Reviewer’s Evaluation |
Response and Revisions |
|
Does the introduction provide sufficient background and include all relevant references? |
Yes |
Additional definitions and structure description added. |
|
Is the research design appropriate? |
Yes |
No changes to design. |
|
Are the methods adequately described? |
Can be improved |
Clarified sample size, added SVI definition, provided references and justifications, reformulated unclear sentences. |
|
Are the results clearly presented? |
Can be improved |
Improved integration of tables/figures with text; clarified 'data not shown'; added explanatory text to Figure 1. |
|
Are the conclusions supported by the results? |
Yes |
Clarified conclusion and adjusted structure to include study limitations and future research recommendations. |
|
Are all figures and tables clear and well-presented? |
Must be improved |
Titles repositioned, explanatory captions improved, Box 1 transformed into a table for clarity. |
Point-by-point Response to Reviewer’s Specific Comments
Definition of social vulnerability (line 74)
Response: A concise definition of 'social vulnerability' was added immediately after its first mention, referencing the literature and clarifying its meaning in the context of food environment studies (Introduction: lines 72 - 75)
Provide the structure of the paper at the end of this section
Response: Two sentences describing the organization of the paper were added to the end of the Introduction (Introduction: lines 80 - 86)
Line 50–51: Provide reference
Response: An appropriate reference was inserted to support the statement (line 49)
Provide definition/description of SVI and meaning of average classification (line 90–92)
Response: A brief definition of SVI, its dimensions, and the interpretation of the average classification (0.33) were added, with relevant references (Methods: lines 98 - 102)
Sample size clarity
Response: The exact sample size (n = 18 schools) was explicitly stated before the sentence discussing statistical precision. (Lines 114 - 119)
Number all equations
Response: All equations in the manuscript have been numbered for clarity.
Line 95–97 unclear
Response: The sentence was reformulated for clarity and grammatical accuracy.
Provide references for lines 108–119
Response: References were added to support the methodological approach described (line 123)
Justification for strict identification/classification steps (lines 118–121)
Response: A justification was added, highlighting that these steps minimized the risk of omitting eligible retail establishments, supported by references to similar approaches in previous studies. (lines 134 - 142)
Line 123 – remove 'The' and reframe
Response: The sentence was reformulated and the unnecessary 'The' removed.
Lines 141 and 144 – 'according' repeated
Response: Sentence reworded to remove repetition and improve clarity.
Box 1 – use a Table
Response: Box 1 content was reformatted into a table (table 1) for improved readability.
Table 1 – lines 295–303 to be placed under table
Response: The relevant sentences were moved to directly follow Table 1.
Figure 1 title placement and lack of explanatory text
Response: Figure 1 title was positioned above the figure, and additional explanatory text was added in the Results section to clearly describe and interpret the map. (Lines 342 - 352)
Table 2 – same comment as Table 1
Response: Captions and related text were repositioned for consistency.
Clarify 'data not shown' (line 344–345)
Response: We thank the reviewer for requesting clarification. The data referred to in this sentence were not presented in a table due to the journal’s limit on the number of tables. To avoid confusion, we have removed the expression “data not shown” from the text in the revised version.
Duplication of text (lines 448–460 and 461–473)
Response: The duplicate section was removed, and in-text citations were verified for accuracy.
Line 495–497 – clarify policy analysis statement
Response: The sentence was revised to clarify that the analysis referenced is from previous literature, not from the authors’ own data (lines 544 - 547)
Move limitations and future research to Conclusion
Response: Limitations and future research recommendations were relocated to the Conclusion section, as suggested.
Proofreading required
Response: Comprehensive proofreading was performed, correcting grammar, removing redundancies, and improving clarity throughout.
Conclusion section update
Response: Conclusion section now includes study limitations and future research, as recommended.
Response to Comments on the Quality of English Language
The manuscript underwent a final proofreading by a professional translator and native English speaker, as certified in the attached English Certification (ID: 92104.297-276, August 14, 2025). This review focused on improving clarity, grammar, and consistency throughout the text.
Changes include:
- Simplifying complex sentences.
- Removing redundancies.
- Ensuring consistency of terminology and verb tense.
- Aligning spelling and style with journal guidelines.
All these minor revisions are highlighted in track changes in the re-submitted manuscript.
This manuscript is a resubmission of an earlier submission. The following is a list of the peer review reports and author responses from that submission.
Round 1
Reviewer 1 Report
Comments and Suggestions for Authors
- The originality of the research is not clear. The authors stated “challenges in the standardization of the methods employed may limit the comparability and applicability of the results (line 58)”, however, the paper did not address this point in the analysis and the method used in the current paper did not improve the comparability and applicability of the results. Investigating a widely discussed topic using existing method in a new context did not constitute enough academic attribution as a research paper. The authors should develop a comprehensive literature review, identify research gaps and establish clear research questions to strengthen the scholarly value of the study.
- The definition and categorization of food environment are problematic and oversimplified. It is not convincing to treat all restaurants and street food as healthy food resource, as in the literature, fast food and full service restaurants are usually treated differently. It is also crucial to investigate the food consumption behavior of the students. For instance, most students will probably not do grocery shopping independently, in which case, to include the supermarket and grocery stores in the study not be meaningful, if they mostly serve the surrounding neighborhood rather than the school students.
- The authors should further explain the sampling process and the data source for food outlets. How were the sample schools selected? Were the 18 schools representative of the 346 schools in the region? This will affect the validity and generalizability of the research conclusion. The authors can consider using POI data to more comprehensively reflect the food environment in the study area.
- It is confusing that in section 2.3.2. “Characterization of the Internal School Environment”, the authors talked about the food marketed in the neighborhood (Line 151) or surrounding the schools (Line 205), does this belong to internal school environment? Should these be discussed in section 2.3.1 (food environment surrounding the school)?
- The authors used too many abbreviations, which makes the article not very readable.
- The format of Table 1 should be refined.
Author Response
Summary
Thank you very much for taking the time to review this manuscript. Please find the detailed responses below and the corresponding revisions/corrections highlighted/in track changes in the re-submitted files.
We opted to provide point-by-point responses to the reviewer’s comments directly in the box below. For convenience, we have also uploaded the same responses as a Word document attachment.
Point-by-point response to Comments and Suggestions for Authors
Comment 1: The originality of the research is not clear. The authors stated “challenges in the standardization of the methods employed may limit the comparability and applicability of the results (line 58)”, however, the paper did not address this point in the analysis and the method used in the current paper did not improve the comparability and applicability of the results. Investigating a widely discussed topic using existing method in a new context did not constitute enough academic attribution as a research paper. The authors should develop a comprehensive literature review, identify research gaps and establish clear research questions to strengthen the scholarly value of the study.
Response 1: We thank the reviewer for the detailed comment. We recognize the importance of making the academic contribution and originality of the study more explicit. To address this point, we made significant adjustments to the Introduction (paragraphs 5, linhas 65 a 72).
“The lack of studies on school food environments is especially concerning in the Federal District (DF), located in Brazil’s Central-West region and home to the nation’s capital. Although Brasília’s central area was planned, the DF is among the most income-segregated regions in the country, with pronounced inequalities across its administrative areas [13,14]. Socioeconomic disparities shape food environments across different territories [15,16]. Thus, understanding food availability and access within and around schools is fundamental for developing effective strategies to promote the health of children and adolescents. “
We reinforced the literature gaps, especially the lack of studies on school food environments in the Federal District (DF) — a region marked by extreme socio-spatial inequalities and limited availability of systematized data on this topic. Additionally, we explicitly included the research question at the end of the introduction (paragraph 6, lines 73–79), as suggested.
To address this gap, this study investigates the following research question: How is the food environment configured inside and around public and private schools in the DF, and how is this configuration associated with social vulnerability? To answer this question, we mapped the surroundings and characterized the internal food environment of public and private schools in the DF. We considered school type and the degree of social vulnerability in each region. This study provides novel, locally grounded evidence to guide more effective and context-sensitive school food policies in Brazil.
Regarding the methodological contribution, it is presented in the Methods section, where we describe the adoption of a classification of food establishments based on the population’s purchasing profile and the principles of the Brazilian Dietary Guidelines — a methodology recently applied in a national study conducted by the Ministry of Social Development and Fight Against Hunger (lines 148–158). This approach represents an important advancement, as it allows for greater comparability across studies and improved applicability of the findings for public policy planning.
In addition, the manuscript presents the conceptual framework used and applies a validated tool to assess the healthiness of canteens and food outlets surrounding schools, which strengthens the analytical rigor and contributes to the originality of the proposed approach.
Comment 2: The definition and categorization of food environment are problematic and oversimplified. It is not convincing to treat all restaurants and street food as healthy food resource, as in the literature, fast food and full service restaurants are usually treated differently. It is also crucial to investigate the food consumption behavior of the students. For instance, most students will probably not do grocery shopping independently, in which case, to include the supermarket and grocery stores in the study not be meaningful, if they mostly serve the surrounding neighborhood rather than the school students.
Response 2: We thank the reviewer for the relevant observations. We agree that the categorization of food outlets requires caution and nuance, and we appreciate the opportunity to clarify our approach.
In our study, establishments were not classified solely based on their commercial type (e.g., restaurant, supermarket, farmers’ market, or street vendor), but rather according to the average food acquisition profile of the population in these locations, as identified in the Brazilian Household Budget Survey (POF) 2017/2018. This methodology was developed by the Ministry of Social Development and Fight Against Hunger (MDS). These sources allowed us to estimate, by state, the average proportion of fresh/minimally processed, processed, and ultra-processed foods acquired at each type of establishment.
Thus, establishments such as restaurants, snack bars, or street vendors were not treated homogeneously, but were classified as “healthy” or “unhealthy” based on the predominance of foods acquired, following the NOVA classification and the recommendations of the Brazilian Dietary Guidelines.
Furthermore, we highlight that the methodology developed by the MDS is considered innovative precisely because it reflects the Brazilian local context, relying on empirical acquisition data per establishment type in each state. In the specific case of the Federal District—where our study was conducted—establishments categorized as “healthy” showed a predominantly healthy sales profile, according to POF 2017/2018 data and the criteria set forth in the Brazilian Dietary Guidelines. Therefore, the grouping of establishments in our study was not arbitrary, but grounded in regionally valid evidence reflecting the actual food environment.
“The establishments were subsequently classified according to the NOVA food classification system [26], according to the methodology proposed in the study by the Brazilian Ministry of Development and Social Assistance, Family and Fight against Hunger (MDS – Ministério do Desenvolvimento e Assistência Social, Família e Combate à Fome) [27]. This methodology is based on data from the 2017/2018 Household Budget Survey (POF) [28], which allowed the identification of the types of food products actually purchased by consumers in each commercial establishment, stratified by Federative Unit.
The classification of establishments reflects the average consumer purchase profile, rather than the store's intended product offering. Based on this information and guided by the Brazilian Dietary Guidelines [29], establishments were grouped into five categories (…)” (lines 145 – 150)
Regarding the absence of data on students’ eating behavior, we acknowledge this as a limitation and have addressed it in the Discussion section. However, the primary objective of this study was to characterize the external school food environment, which can influence food behavior through availability, exposure, and social context.
Additionally, international evidence shows that adolescents do consume food in the school surroundings, including from supermarkets, which supports the inclusion of these establishments. In the article by Macdiarmid JI et al, Supermarkets were the outlets from which pupils reported most often making purchases. (Macdiarmid JI, Wills WJ, Masson LF, Craig LCA, Bromley C, McNeill G. Food and drink purchasing habits out of school at lunchtime: a national survey of secondary school pupils in Scotland. Int J Behav Nutr Phys Act 2015 Aug 4:12:98. doi: 10.1186/s12966-015-0259-4).
Comment 3: The authors should further explain the sampling process and the data source for food outlets. How were the sample schools selected? Were the 18 schools representative of the 346 schools in the region? This will affect the validity and generalizability of the research conclusion. The authors can consider using POI data to more comprehensively reflect the food environment in the study area.
Response 3: Thank you for this valuable and constructive comment. We have revised the manuscript to clarify both the school sampling strategy and the data sources used to characterize the food environment.
Regarding the sampling process, the study employed a probabilistic sampling design stratified by school administrative type (public/private). The sampling frame consisted of all 346 urban schools in the Federal District that offered ninth-grade classes, according to the 2021 School Census. A total of 20 schools were initially selected (9 public and 11 private). After two refusals from private schools, the final sample comprised 18 schools, equally distributed between public and private networks.
While the relatively small sample size may limit statistical precision, the probabilistic nature of the sampling design ensures representativeness of the target population. As established in sampling theory, representativeness is a property of how the sample is selected, whereas sample size primarily affects the margin of error and the power to detect significant differences.
This descriptive ecological study was carried out with urban schools from the public and private networks of the DF, selected by probabilistic sampling stratified by type of school. The study population corresponds to all the urban schools in the DF that have ninth grade, totaling 346 schools (159 public and 187 private) [19] to compare with the Brazilian National Survey of School Health (PeNSE – Pesquisa Nacional de Saúde do Escolar) [20]. Although the small sample size may limit the statistical precision of the estimates, the probabilistic design ensures that the sample is representative of the target population. (lines 93 – 100)
Regarding the data source for food outlets, we used the Annual List of Social Information (RAIS – Relação Anual de Informações Sociais), an official administrative database provided by the Brazilian Ministry of Labor and Employment. The 2021 RAIS was used to identify formal food retail and service establishments located within an 800-meter radius of each sampled school. Establishments were selected based on 12 CNAE codes (National Classification of Economic Activities), consistent with the literature on school food environments and aligned with the national food desert mapping methodology developed by CAISAN (2018).
To address limitations inherent to RAIS—such as the exclusion of informal vendors and potential classification errors—we conducted a virtual audit using Google Street View (2023). This allowed us to validate the existence and location of establishments, correct CNAE codes when their registered activity did not match the storefront, and include additional eligible establishments not captured in the RAIS. All verified food outlets were then georeferenced in QGIS, with the creation of 250 m and 400 m buffers around each school to ensure consistent spatial analysis.
We acknowledge that there are some limitations in the adopted methodology, as outlined in the following section.
Limitations of this study include the absence of on-site data on informal street vendors, whose presence was reported by school representatives. Given the informal and dynamic nature of this type of commerce, students’ real exposure to these food sources was probably underestimated. Additionally, the use of administrative data (RAIS) and virtual tools (Google Street View) may not fully reflect current operations or food offerings at the time of student access, although a detailed virtual audit was conducted to verify the existence, location, and façade of establishments, including CNAE corrections and inclusion of unlisted eligible outlets. (lines 508 – 515)
We appreciate the reviewer’s suggestion to consider the use of Points of Interest (POI) data. However, we believe that the virtual audit strategy adopted in this study fulfills a similar role, offering the added benefit of direct visual verification and manual correction of classification errors that often affect automated POI datasets. In this sense, we argue that our approach may offer greater methodological robustness. Nevertheless, we acknowledge the limitations of using RAIS alone—particularly the lack of informal vendors—and have highlighted this point in the Discussion as an area for improvement in future research, including the possible integration of POI-based datasets.
Comment 4: It is confusing that in section 2.3.2. “Characterization of the Internal School Environment”, the authors talked about the food marketed in the neighborhood (Line 151) or surrounding the schools (Line 205), does this belong to internal school environment? Should these be discussed in section 2.3.1 (food environment surrounding the school)?
Response 4: Thank you for your insightful comment. To improve clarity and reflect the methodological distinctions between the two types of data collected, we reorganized the manuscript structure.
In the Methods section, we divided the content into:
• 2.3.1. Characterization of the School Surroundings: Data Sources and Classification Methods, which addresses the objective geospatial mapping of food outlets using secondary data (RAIS, Google Street View, QGIS); and
• 2.3.2. Characterization of the School Food Environment: Data Collection from School Staff, which presents data from school staff reports about both internal food practices and their perception of the surrounding environment.
In the Results section, the data were similarly presented in two parts:
• 3.1. Characterization of the School Surroundings, based on georeferenced mapping; and
• 3.2. Internal and Surrounding Food Environment According to School Staff Reports, reflecting the subjective assessment of principals and pedagogical coordinators.
This distinction ensures consistency and allows us to respect the methodological differences between objective and reported data. We believe this reorganization clarifies the scope of each subsection and addresses the reviewer’s concern regarding potential confusion.
Comment 5: The authors used too many abbreviations, which makes the article not very readable.
Response 5: Thank you for this valuable feedback. We agree that excessive use of abbreviations can hinder readability, especially for interdisciplinary or international audiences. In response, we thoroughly revised the manuscript to reduce the number of abbreviations and to prioritize clarity over conciseness.
Comment 6: The format of Table 1 should be refined.
Response 6: Thank you for your helpful suggestion. To improve the clarity and visual organization of Table 1, we revised its structure and content. Specifically, we removed the column “Schools with at least 1 establishment – N; %”, which concentrated contextual information that was better suited for narrative explanation. The most relevant findings previously included in that column are now described in the text of the Results section, allowing for a more fluid and reader-friendly presentation.
We also adjusted the column layout and labels to improve alignment, consistency, and readability. These modifications aim to enhance the overall presentation of the table and support clearer interpretation of the data.
Response to Comments on the Quality of English Language
The English in this article has been carefully revised with a focus on clarity, sentence structure, and academic style. The current version avoids long and complex sentences and ensures coherence across paragraphs. Furthermore, the manuscript has been professionally reviewed and proofread by a native English speaker with expertise in academic writing.
Reviewer 2 Report
Comments and Suggestions for Authors
The article presents an analysis of the food environment in public and private schools in the Federal District of Brazil. The study focuses on the availability of healthy and unhealthy food products within the schools and their surroundings. In my opinion, the publication is a very valuable work that addresses a current and socially significant topic — the growing problem of obesity and unhealthy eating among children and adolescents. Furthermore, the study employs a solid methodology, using the NOVA classification and differentiated buffer zones (250 m, 400 m, and 800 m), which provides a more in-depth picture of food availability. An additional strength is the comparison based on school type and level of social vulnerability, which allows the identification of inequalities in the food environment. Finally, the article includes concrete recommendations regarding legislative changes and implementation measures.
My suggestions for improvement:
- Include a qualitative analysis on students’ purchasing motivations, as there is a lack of information about students’ own food preferences and decision-making. This limits the interpretation of how the environment influences behavior.
- Consider food seasonality. The study is cross-sectional, and the food environment may vary with the seasons or over time (e.g., temporary food vendors appearing near schools).
- In future research, data should be collected across more than one time period (e.g., before and after the implementation of regulations).
These suggestions might form the basis for the content of a future article by the same research team.
4. Linguistic and syntactic issues. There are some language-related problems in the text (e.g., complex, unclear sentences and excessively long paragraphs), which hinder the readability and flow of the article.
Author Response
Summary
Thank you very much for taking the time to review this manuscript. Please find the detailed responses below and the corresponding revisions/corrections highlighted/in track changes in the re-submitted files.
We opted to provide point-by-point responses to the reviewer’s comments directly in the box below. For convenience, we have also uploaded the same responses as a Word document attachment.
Point-by-point response to Comments and Suggestions for Authors
General Comment: The article presents an analysis of the food environment in public and private schools in the Federal District of Brazil. The study focuses on the availability of healthy and unhealthy food products within the schools and their surroundings. In my opinion, the publication is a very valuable work that addresses a current and socially significant topic — the growing problem of obesity and unhealthy eating among children and adolescents. Furthermore, the study employs a solid methodology, using the NOVA classification and differentiated buffer zones (250 m, 400 m, and 800 m), which provides a more in-depth picture of food availability. An additional strength is the comparison based on school type and level of social vulnerability, which allows the identification of inequalities in the food environment. Finally, the article includes concrete recommendations regarding legislative changes and implementation measures.
Response: We sincerely thank Reviewer 2 for the thoughtful and encouraging feedback. We greatly appreciate your recognition of the relevance and originality of our work, as well as your positive assessment of the methodological rigor, including the use of the NOVA classification, the differentiated buffer zones (250 m, 400 m, and 800 m), and the comparative analysis by school type and social vulnerability.
We are especially grateful for your comments on the social relevance of the topic and on the value of the policy recommendations included in the manuscript. Although no specific changes were requested, your feedback reinforced the clarity of our research design and the importance of communicating our findings effectively. We were motivated to review the final version carefully to ensure that these strengths are well-reflected throughout the text.
Thank you once again for your valuable contribution to the improvement of our manuscript.
Comment 1:
Include a qualitative analysis on students’ purchasing motivations, as there is a lack of information about students’ own food preferences and decision-making. This limits the interpretation of how the environment influences behavior.
Response 1:
We thank the reviewer for this pertinent suggestion. We agree that understanding students’ food preferences and purchasing motivations would greatly enrich the analysis by providing a deeper perspective on how the school food environment influences behavior. Unfortunately, such data were not available in the present study.
However, we clarify that the present study employed an ecological design, focusing on the objective characterization of internal and external school food environments, without collecting individual-level data from students. Therefore, qualitative information on adolescents’ perceptions and motivations lies beyond the scope of this specific publication.
We recognize this as a limitation and have added a paragraph in the Discussion section addressing the absence of behavioral and attitudinal data, and how this may influence the interpretation of our findings.
The study also did not assess students’ actual consumption or preferences, which limits interpretation of how the food environment affects behaviors. Investigating students’ purchasing motivations and decision-making would provide important context and is recommended for future studies. (lines 522 – 525)
Comment 2:
Consider food seasonality. The study is cross-sectional, and the food environment may vary with the seasons or over time (e.g., temporary food vendors appearing near schools).
Comment 3: In future research, data should be collected across more than one time period (e.g., before and after the implementation of regulations).
Response to comments 2 and 3:
We thank the reviewer for this important observation. Indeed, we recognize that the cross-sectional design of the study limits our ability to capture seasonal or temporal variations in the food environment, such as the presence of temporary or informal food vendors that may fluctuate throughout the year or across different days of the week.
To mitigate this limitation, we used a virtual audit based on updated Google Street View images, which allowed us to verify the physical presence and storefronts of food establishments as close as possible to the time of data collection. However, we acknowledge that this strategy does not fully capture the informal market or temporary establishments, which may be relevant to students' food consumption.
This limitation has been explicitly acknowledged in the Discussion section, where we reflect on its potential impact on the characterization of the food environment and suggest future research using longitudinal designs or in-person observations to better capture these dynamics.
Furthermore, the cross-sectional nature of the study limits the ability to account for temporal variations, such as the presence of seasonal food vendors or changes in food offerings throughout the school year. The collection was conducted at a single point in time, which may not capture fluctuations in the food environment. Data collection at multiple time points is recommended for future research, especially before and after implementation of new food regulations. (lines 526 – 531).
Comment 4: Linguistic and syntactic issues. There are some language-related problems in the text (e.g., complex, unclear sentences and excessively long paragraphs), which hinder the readability and flow of the article.
Response 4: Thank you for this important observation. In response to your comment, we conducted a careful revision of the manuscript to improve linguistic clarity, sentence structure, and overall readability.
We simplified complex sentences, reduced paragraph length where appropriate, and ensured greater coherence across sections. Additionally, the revised manuscript was professionally proofread by a native English speaker with experience in academic editing, to ensure a high standard of language quality throughout the text.
We believe that these revisions significantly enhance the flow and accessibility of the article for a broad academic audience.
Reviewer 3 Report
Comments and Suggestions for Authors
Major Comments
- The sample is not representative. The study includes only 18 schools (9 public and 9 private), which is quite limited for robust conclusion. Two private schools declined to participate, which might introduce some bias.
- The presence of informal vendors around schools was based on school staff reports rather than on-site observation. Given how common and variable informal food sales can be, relying on second-hand reports underestimates students' real exposure to these food sources.
- Reliance on secondary and online data․ The classification of food retailers was largely done using administrative data and Google Street View, which may not accurately reflect current operations or food offerings. This limits the reliability of the food environment mapping.
- Accessibility issues. The use of straight-line (Euclidean) distances to define 250 m, 400 m, and 800 m buffers may not realistically reflect how students actually access food outlets. Urban infrastructure, road safety, and walking paths could greatly affect accessibility but are not accounted for.
- Categorizing outlets as either “healthy” or “unhealthy” is not justificated scientifically. Many shops sell both ultra-processed and fresh items, and this nuance is important when analyzing food choices and policy implications.
- The study offers a good overview from school administrators’ perspective, but hearing directly from students or parents about what foods are bought and consumed would have added valuable context to the findings.
- While the paper does a thorough job of describing what is available, it doesn’t explore whether students are actually consuming these foods or whether it relates to their health status. 8. The study notes that some schools aren’t complying with regulations on food sales. A deeper reflection on the barriers to implementing these policies would improve the practical value of the work.
Minor Comments
The spatial figure mentioned (Figure 1) could be more visually engaging by distinguishing schools by type or highlighting establishment types more clearly.
Suggestions for improvement
Expand on why current food regulations are not being followed in private chools and suggest how enforcement could be improved.
If possible, include some form of direct observation or field verification, especially for informal vendors. Consider using network-based distances or, at minimum, explain why Euclidean buffers were chosen despite their limitations.
Even a small number of interviews with students or parents would enrich the findings. A short paragraph reflecting on how the findings might or might not apply to other parts of Brazil would help international readers place this in context.
Author Response
Summary
Thank you very much for taking the time to review this manuscript. Please find the detailed responses below and the corresponding revisions/corrections highlighted/in track changes in the re-submitted files.
We opted to provide point-by-point responses to the reviewer’s comments directly in the box below. For convenience, we have also uploaded the same responses as a Word document attachment.
Point-by-point response to Comments and Suggestions for Authors
Comment 1: The sample is not representative. The study includes only 18 schools (9 public and 9 private), which is quite limited for robust conclusion. Two private schools declined to participate, which might introduce some bias.
Response 1: We thank the reviewer for this thoughtful observation. We agree that the small sample size is a limitation that may affect the precision and power of the analyses, particularly in detecting statistically significant differences.
However, we clarify that the study used a probabilistic sampling strategy, stratified by school administrative type (public/private), based on the complete universe of 346 eligible urban schools with 9th-grade classes in the Federal District (2021 School Census). A total of 20 schools were initially selected, and two private schools declined to participate. The final sample included 18 schools (9 public and 9 private).
(…) However, two private schools refused to participate, resulting in a final sample of 18 schools – nine public and nine private – providing an equal allocation between the types. The reasons for refusal were not related to school characteristics. As in most probability samples, when there is no available data on non-respondents, we assumed that no systematic differences exist between participants and non-participants. This assumption minimized the risk of selection bias and supported the validity of the sample design (lines 105 – 111)
While sample size affects the margin of error, representativeness is a function of how the sample is selected, as defined by sampling theory. Because the sample was randomly selected from a defined population, it retains its representativeness despite its reduced size.
That said, we acknowledge the potential nonresponse bias introduced by the two private schools that declined participation. This limitation is now discussed in the revised Discussion section, where we also reflect on how it may affect generalizability and encourage caution in interpreting the findings.
Although two private schools declined participation, their refusal did not appear to be associated with any identifiable institutional characteristics. Nonetheless, as with most probability samples, it was not possible to compare respondents and non-respondents directly. Thus, we assumed that no systematic differences exist between these groups. While this assumption is methodologically acceptable, it introduces a potential, albeit limited, source of selection bias that should be acknowledged. (lines 432 – 437).
Comment 2: The presence of informal vendors around schools was based on school staff reports rather than on-site observation. Given how common and variable informal food sales can be, relying on second-hand reports underestimates students' real exposure to these food sources.
Response 2: Thank you for this important observation. We agree that informal food vendors play a significant and often underestimated role in the school food environment. In our study, information about informal vendors was indeed collected through reports by school staff, and no systematic on-site observation of these vendors was conducted.
We recognize that this approach may lead to underestimation of students’ real exposure, given the dynamic, mobile, and unregulated nature of informal food sales around schools. This limitation has been explicitly addressed in the revised Discussion section, where we acknowledge the absence of direct field data on informal vendors and reflect on how this may affect the accuracy of the characterization of the food environment.
We also highlight that even though we used Google Street View and administrative data (RAIS) to map formal establishments, these tools have inherent limitations in capturing informal or temporary commerce. As such, we have recommended that future studies include on-site observations or mixed methods approaches to better document these dynamic food sources.
Limitations of this study include the absence of on-site data on informal street vendors, whose presence was reported by school representatives. Given the informal and dynamic nature of this type of commerce, students’ real exposure to these food sources was probably underestimated. Additionally, the use of administrative data (RAIS) and virtual tools (Google Street View) may not fully reflect current operations or food offerings at the time of student access, although a detailed virtual audit was conducted to verify the existence, location, and façade of establishments, including CNAE corrections and inclusion of unlisted eligible outlets. (lines 508 – 511)
Comment 3: Reliance on secondary and online data. The classification of food retailers was largely done using administrative data and Google Street View, which may not accurately reflect current operations or food offerings. This limits the reliability of the food environment mapping.
Response 3:
Thank you for this important observation. We agree that reliance on secondary and virtual sources may limit the accuracy of food environment mapping, particularly in rapidly changing or informal contexts.
To address this, we conducted a detailed virtual audit using Google Street View images, which allowed us to verify the existence, location, and façade of each establishment, correct misclassified CNAE codes, and include additional eligible outlets not listed in the administrative database (RAIS). This method has been increasingly adopted in international and Brazilian studies to characterize the school food environment and validate secondary data sources [1–4]. Although this approach improves accuracy compared to the use of administrative data alone, we acknowledge that it may still fail to capture recent openings, closings, or informal vendors.
This limitation has been explicitly discussed in the revised Discussion section, where we reflect on the potential impact on data reliability and suggest that future research incorporate in-person field audits or mixed-methods approaches to validate and enrich secondary data sources.
Limitations of this study include the absence of on-site data on informal street vendors, whose presence was reported by school representatives. Given the informal and dynamic nature of this type of commerce, students’ real exposure to these food sources was probably underestimated. Additionally, the use of administrative data (RAIS) and virtual tools (Google Street View) may not fully reflect current operations or food offerings at the time of student access, although a detailed virtual audit was conducted to verify the existence, location, and façade of establishments, including CNAE corrections and inclusion of unlisted eligible outlets. (lines 508 – 511)
References
1. Tse K;Zeng MX;Gibson AA;Partridge SR;Raeside R;Valanju R;et al. Retrospective analysis of regional and metropolitan school food environments using Google Street View: A case study in New South Wales, Australia with youth consultation. Heal Promot J Aust. 2024;(February 2024):1–11.
2. da Costa Peres CM;de Lima Costa BV;Pessoa MC;Honório OS;do Carmo AS;da Silva TPR;et al. Community food environment and presence of food swamps around schools in a Brazilian metropolis. Cad Saude Publica. 2021;37(5).
3. Lake AA;Burgoine T;Stamp E;Grieve R. The foodscape: Classification and field validation of secondary data sources across urban/rural and socio-economic classifications in England. Int J Behav Nutr Phys Act [Internet]. 2012;9(1):37. Available from: http://www.ijbnpa.org/content/9/1/37
4. Huang D;Brien A;Omari L;Culpin A;Smith M;Egli V. Bus Stops near Schools Advertising Junk Food and Sugary Drinks. Nutrients. 2020;12:1–19.
Comment 4: Accessibility issues. The use of straight-line (Euclidean) distances to define 250 m, 400 m, and 800 m buffers may not realistically reflect how students actually access food outlets. Urban infrastructure, road safety, and walking paths could greatly affect accessibility but are not accounted for.
Response 4:
Thank you for this insightful comment. We agree that Euclidean (straight-line) buffers do not capture the full complexity of real-world accessibility, which may be influenced by urban infrastructure, road safety, pedestrian routes, and physical barriers.
However, the use of Euclidean buffers remains a widely accepted and commonly used method in international studies on food environments, particularly when detailed pedestrian network data are unavailable or inconsistent. Several studies have adopted fixed-radius Euclidean buffers of 250 m [2], 400m [5,6], and 800m [7–9] to assess students' exposure to food retailers around schools, supporting the methodological choice made in the present study. This approach allows for comparability across studies and contributes to methodological transparency and replicability.
We acknowledge this limitation in the revised Discussion section, where we reflect on how using network-based measures could provide more accurate insights in future research. Despite its limitations, the Euclidean approach still provides a useful and systematic proxy for potential food exposure in the school surroundings.
The use of Euclidean (straight-line) distances to define the 250 m, 400 m, and 800 m buffers may not correspond to the actual routes students use, especially in urban areas with barriers such as highways, a lack of sidewalks, or unsafe crossings. Although this technique is commonly used in geographic studies due to its simplicity and comparability, future studies could explore the application of network-based distances to improve precision. (lines 516 – 521).
References
2. da Costa Peres CM;de Lima Costa BV;Pessoa MC;Honório OS;do Carmo AS;da Silva TPR;et al. Community food environment and presence of food swamps around schools in a Brazilian metropolis. Cad Saude Publica. 2021;37(5).
5. Londoño-Cañola C;Serral G;Díez J;Martínez-García A;Franco M;Artazcoz L;et al. Retail Food Environment around Schools in Barcelona by Neighborhood Socioeconomic Status: Implications for Local Food Policy. Int J Environ Res Public Health. 2023;20(1).
6. Corrêa EN;Rossi CE;Das Neves J;Silva DAS;De Vasconcelos FDAG. Utilization and environmental availability of food outlets and overweight/obesity among schoolchildren in a city in the south of Brazil. J Public Health (Bangkok). 2018;40(1):106–13.
7. Howard PH;Fitzpatrick M;Fulfrost B. Proximity of food retailers to schools and rates of overweight ninth grade students: An ecological study in California. BMC Public Health. 2011;11.
8. Assis MM;Gratão LHA;da Silva TPR;Cordeiro NG;do Carmo AS;de Freitas Cunha C;et al. School environment and obesity in adolescents from a Brazilian metropolis: cross-sectional study. BMC Public Health. 2022;22(1):2–11.
9. França FCO de;Andrade I da S;Zandonadi RP;Sávio KE;Akutsu R de CC de A. Food Environment around Schools: A Systematic Scope Review. Nutrients. 2022;14(23):2–17.
Comment 5: Categorizing outlets as either “healthy” or “unhealthy” is not justificated scientifically. Many shops sell both ultra-processed and fresh items, and this nuance is important when analyzing food choices and policy implications.
Response 5:
Thank you for raising this important point. We fully agree that food environments are complex and that many outlets offer a mixed variety of products, including both ultra-processed and fresh/minimally processed items.
To address this complexity, our study adopted a context-sensitive classification system based not solely on commercial categories, but on the predominant sales profile of each outlet type, as identified in the Brazilian Household Budget Survey (POF 2017/2018). This classification was developed and applied in a national study by the Ministry of Social Development (MDS) and considers the actual food acquisition patterns of the population — rather than formal retail definitions — following the principles of the Brazilian Dietary Guidelines and the NOVA classification.
Rather than assuming that all outlets of a given type (e.g., bakeries, markets, snack bars) are either healthy or unhealthy, we grouped establishments according to the proportion of fresh/minimally processed foods versus ultra-processed foods they typically sell. Mixed categories were also included (e.g., “mixed fresh,” “mixed processed,” and “other mixed”) to reflect the gradient of food availability and avoid overly binary categorizations.
This approach is described in detail in the Methods section (lines 141 – 169), and we believe it offers a more nuanced, evidence-based, and policy-relevant perspective on food outlet classification.
Comment 6: The study offers a good overview from school administrators’ perspective, but hearing directly from students or parents about what foods are bought and consumed would have added valuable context to the findings.
Comment 7: While the paper does a thorough job of describing what is available, it doesn’t explore whether students are actually consuming these foods or whether it relates to their health status.
Response to Comments 6 and 7:
Thank you for these thoughtful and important observations. We agree that incorporating first-hand accounts from students and parents, as well as data on actual food consumption and health outcomes, would provide valuable context to better understand how the school food environment influences behaviors and nutritional status.
However, we clarify that the present study employed an ecological design, focusing on the objective characterization of internal and external school food environments, without collecting individual-level data from students. Qualitative information on students’ perceptions, purchasing motivations, and family influences, as well as quantitative data on food consumption or anthropometric indicators, were beyond the scope of this specific publication.
The study prioritized the institutional perspective, gathering information from school principals and pedagogical coordinators to understand how the food environment is structured and perceived at the administrative level. We recognize that this approach does not capture student or family behaviors directly, and this limitation has been acknowledged in the Discussion section.
The study also did not assess students’ actual consumption or preferences, which limits interpretation of how the food environment affects behaviors. Investigating students’ purchasing motivations and decision-making would provide important context and is recommended for future studies. (lines 522 – 525)
Comment 8: The study notes that some schools aren’t complying with regulations on food sales. A deeper reflection on the barriers to implementing these policies would improve the practical value of the work.
Response 8:
Thank you for this thoughtful and relevant suggestion. We agree that reflecting on the barriers to implementing school food regulations is essential to increase the practical and policy value of the study.
Our data collection identified instances where schools were not fully compliant with existing regulations. However, the study did not include specific qualitative methods (e.g., interviews or focus groups) aimed at exploring the underlying reasons for non-compliance, such as lack of oversight, economic pressures, or institutional challenges. Therefore, we are limited in our ability to draw definitive conclusions on this aspect.
Nonetheless, we expanded the Discussion section to include a reflection on potential barriers to effective policy implementation, drawing from relevant literature and insights reported by school administrators. These include insufficient enforcement, economic incentives for canteen operators, lack of training, and the absence of coordinated intersectoral support.
The implementation of school food regulations faces significant challenges, particularly in private institutions. These schools often show greater resistance to compliance due to weaker institutional structures and the absence of public initiatives such as the National School Feeding Program (PNAE), which facilitates adherence in public schools [10,11]. The effectiveness of regulatory measures is further undermined by the lack of educational campaigns, poor intersectoral coordination, political and economic pressures from the private sector, and inadequate monitoring systems [11]. In the Federal District, although Law No. 5,321/2014 assigns the Health Department responsibility for inspecting school canteens [12], no systematic mechanisms for monitoring and enforcement have been effectively implemented. Although these barriers were not directly assessed in the present study, they are supported by previous research and provide important context for understanding the limited enforcement and compliance observed in some schools [10,11,13]. (lines 456 – 468)
We agree that future studies should explore this issue in more depth, using qualitative methods to better understand the institutional, economic, and social factors that hinder effective regulation in school environments.
Comment 9: The spatial figure mentioned (Figure 1) could be more visually engaging by distinguishing schools by type or highlighting establishment types more clearly.
Response 9:
Thank you for this helpful suggestion. In response to your comment, we revised Figure 1 to improve its visual clarity and communicative value. We believe that these adjustments make the figure more engaging and informative for readers and improve the visual presentation of key findings.
Comment 10: Expand on why current food regulations are not being followed in private schools and suggest how enforcement could be improved.
Response 10:
Thank you for this thoughtful suggestion. In the revised Discussion section (lines 456–468 and 490–496), we expanded the reflection on the barriers to implementing school food regulations, particularly in private institutions. These sections now include an evidence-based explanation of the structural and institutional challenges faced by private schools, as well as recommendations to improve enforcement, such as clearer inspection protocols, intersectoral coordination, and capacity-building. Although these aspects were not directly evaluated in the present study, they are supported by previous research and help contextualize the compliance gaps observed in our findings.
Comment 11: If possible, include some form of direct observation or field verification, especially for informal vendors. Consider using network-based distances or, at minimum, explain why Euclidean buffers were chosen despite their limitations.
Response 11:
Thank you for this helpful and integrative comment. We agree that both direct field observation of informal vendors and the use of network-based distance measures would enhance the precision and contextual accuracy of food environment assessments.
However, due to logistical constraints and the study’s ecological design, direct fieldwork was not conducted. To partially address this limitation, we conducted a detailed virtual audit using Google Street View to validate the presence and characteristics of formal establishments. The absence of observational data on informal vendors has been explicitly acknowledged in the revised Discussion section, along with a reflection on its potential impact on exposure estimates.
Limitations of this study include the absence of on-site data on informal street vendors, whose presence was reported by school representatives. Given the informal and dynamic nature of this type of commerce, students’ real exposure to these food sources was probably underestimated. Additionally, the use of administrative data (RAIS) and virtual tools (Google Street View) may not fully reflect current operations or food offerings at the time of student access, although a detailed virtual audit was conducted to verify the existence, location, and façade of establishments, including CNAE corrections and inclusion of unlisted eligible outlets. (lines 510 – 517)
Regarding the use of Euclidean buffers (250 m, 400 m, and 800 m), we adopted this method in alignment with existing literature on school food environments, which frequently applies straight-line distances when detailed pedestrian or road network data are unavailable or inconsistent. Several studies have used fixed-radius Euclidean buffers of 250 m [2], 400m [5,6], 800m [7–9] to estimate food retailer exposure around schools, supporting our methodological choice and ensuring comparability with previous research. This decision and its limitations are also discussed in the manuscript, where we suggest that future research incorporate network-based buffers or walkability measures to improve accuracy.
The use of Euclidean (straight-line) distances to define the 250 m, 400 m, and 800 m buffers may not correspond to the actual routes students use, especially in urban areas with barriers such as highways, a lack of sidewalks, or unsafe crossings. Although this technique is commonly used in geographic studies due to its simplicity and comparability, future studies could explore the application of network-based distances to improve precision. (lines 518 – 523).
Comment 12: Even a small number of interviews with students or parents would enrich the findings. A short paragraph reflecting on how the findings might or might not apply to other parts of Brazil would help international readers place this in context.
Response 12:
Thank you for this thoughtful and constructive comment. We agree that interviews with students or parents would provide valuable insights to complement the school-level data presented here. However, as previously noted, the present study employed an ecological design, and the collection of individual-level data was not included in this specific stage of the research. We recognize this as a limitation and have reflected on it in the Discussion section.
We also appreciate the suggestion to reflect on the transferability of our findings to other regions of Brazil. In response, we added a short paragraph in the Discussion highlighting that while some patterns. We believe this addition helps international readers better situate our findings within the broader Brazilian context.
Finally, although this study was conducted in the Federal District—a region with specific socioeconomic and institutional characteristics—its findings may reflect patterns common to other Brazilian urban centers. However, local policies, urban infrastructure, and school regulation enforcement may vary substantially. Therefore, broader and multicentric studies are needed to confirm and expand upon these findings in diverse contexts. (lines 540 – 544).
4. Response to Comments on the Quality of English Language
The English in this article has been carefully revised with a focus on clarity, sentence structure, and academic style. The current version avoids long and complex sentences and ensures coherence across paragraphs. Furthermore, the manuscript has been professionally reviewed and proofread by a native English speaker with expertise in academic writing.
References used in the responses
1. Tse K;Zeng MX;Gibson AA;Partridge SR;Raeside R;Valanju R;et al. Retrospective analysis of regional and metropolitan school food environments using Google Street View: A case study in New South Wales, Australia with youth consultation. Heal Promot J Aust. 2024;(February 2024):1–11.
2. da Costa Peres CM;de Lima Costa BV;Pessoa MC;Honório OS;do Carmo AS;da Silva TPR;et al. Community food environment and presence of food swamps around schools in a Brazilian metropolis. Cad Saude Publica. 2021;37(5).
3. Lake AA;Burgoine T;Stamp E;Grieve R. The foodscape: Classification and field validation of secondary data sources across urban/rural and socio-economic classifications in England. Int J Behav Nutr Phys Act [Internet]. 2012;9(1):37. Available from: http://www.ijbnpa.org/content/9/1/37
4. Huang D;Brien A;Omari L;Culpin A;Smith M;Egli V. Bus Stops near Schools Advertising Junk Food and Sugary Drinks. Nutrients. 2020;12:1–19.
5. Londoño-Cañola C;Serral G;Díez J;Martínez-García A;Franco M;Artazcoz L;et al. Retail Food Environment around Schools in Barcelona by Neighborhood Socioeconomic Status: Implications for Local Food Policy. Int J Environ Res Public Health. 2023;20(1).
6. Corrêa EN;Rossi CE;Das Neves J;Silva DAS;De Vasconcelos FDAG. Utilization and environmental availability of food outlets and overweight/obesity among schoolchildren in a city in the south of Brazil. J Public Health (Bangkok). 2018;40(1):106–13.
7. Howard PH;Fitzpatrick M;Fulfrost B. Proximity of food retailers to schools and rates of overweight ninth grade students: An ecological study in California. BMC Public Health. 2011;11.
8. Assis MM;Gratão LHA;da Silva TPR;Cordeiro NG;do Carmo AS;de Freitas Cunha C;et al. School environment and obesity in adolescents from a Brazilian metropolis: cross-sectional study. BMC Public Health. 2022;22(1):2–11.
9. França FCO de;Andrade I da S;Zandonadi RP;Sávio KE;Akutsu R de CC de A. Food Environment around Schools: A Systematic Scope Review. Nutrients. 2022;14(23):2–17.
10. Rocha LL;Cordeiro NG;Jardim MZ;Kurihayashi AY;Gentil PC;Russo GC;et al. Do Brazilian regulatory measures promote sustainable and healthy eating in the school food environment? BMC Public Health. 2023;23(1):1–9.
11. Brasil. Ministério da Saúde. Regulamentação da Comercialização de Alimentos em Escolas no Brasil : Experiências estaduais e municipais. Brasília; 2007.
12. Distrito Federal. Governo do Distrito Federal (GDF). DECRETO No 36.900, DE 23 DE NOVEMBRO DE 2015. Brasília; 2015.
13. Micha R;Karageorgou D;Bakogianni I;Trichia E;Whitsel LP;Story M;et al. Effectiveness of school food environment policies on children’s dietary behaviors: A systematic review and meta-analysis. PLoS One. 2018;13(3):1–27.
Reviewer 4 Report
Comments and Suggestions for Authors This is a relevant and well-structured descriptive ecological study that investigates the food environment of public and private schools in the Federal District of Brazil. The topic is particularly important in the context of growing concerns about obesity among adolescents, a population that is especially vulnerable due to the critical developmental stage in which lifelong dietary habits are formed. The study addresses a significant regional gap, as most existing literature focuses on the Southeast of Brazil, and it contributes valuable data to inform local public policies. The stratification by school type and Social Vulnerability Index (SVI), along with the use of multiple geospatial buffers and validated indicators (e.g. Healthiness Index, NOVA classification) adds methodological strength. The comparison between internal and external school food environments further enhances the study’s contribution. However, I would like to address few points that needs more clarification through the manuscript. Firstly, the authors should clarify the central research question earlier in the Introduction. While the objectives are mentioned, stating a clear guiding hypothesis or research question would help focus the narrative. Moreover, although the methods section is generally clear, more details on how the virtual audit was conducted and validated (e.g. explain the use of Google Street View) would increase reliability of the paper. Explain how potential selection bias from schools that declined participation was addressed. According to the conclusion section, private schools are more exposed to unhealthy environments is supported by the evidence, but the policy implications could be better developed. It may also be worth discussing how enforcement of regulations could be improved, especially given that all private schools offered prohibited items. In addition, tables are informative but dense at the same time which may authors could consider highlighting key comparisons in bold or footnotes for easier interpretation. Around the references, they seem adequate but the Discussion section would benefit from more international comparisons (e.g. school food environments in other Latin American or countries with the similar income/condition). The policy section could also cite relevant WHO or FAO guidelines to reinforce the argument for local policy updates."Author Response
Summary
Thank you very much for taking the time to review this manuscript. Please find the detailed responses below and the corresponding revisions/corrections highlighted/in track changes in the re-submitted files.
We opted to provide point-by-point responses to the reviewer’s comments directly in the box below. For convenience, we have also uploaded the same responses as a Word document attachment.
Point-by-point response to Comments and Suggestions for Authors
Comments: This is a relevant and well-structured descriptive ecological study that investigates the food environment of public and private schools in the Federal District of Brazil. The topic is particularly important in the context of growing concerns about obesity among adolescents, a population that is especially vulnerable due to the critical developmental stage in which lifelong dietary habits are formed. The study addresses a significant regional gap, as most existing literature focuses on the Southeast of Brazil, and it contributes valuable data to inform local public policies. The stratification by school type and Social Vulnerability Index (SVI), along with the use of multiple geospatial buffers and validated indicators (e.g. Healthiness Index, NOVA classification) adds methodological strength. The comparison between internal and external school food environments further enhances the study’s contribution. However, I would like to address few points that needs more clarification through the manuscript. Firstly, the authors should clarify the central research question earlier in the Introduction. While the objectives are mentioned, stating a clear guiding hypothesis or research question would help focus the narrative. Moreover, although the methods section is generally clear, more details on how the virtual audit was conducted and validated (e.g. explain the use of Google Street View) would increase reliability of the paper. Explain how potential selection bias from schools that declined participation was addressed. According to the conclusion section, private schools are more exposed to unhealthy environments is supported by the evidence, but the policy implications could be better developed. It may also be worth discussing how enforcement of regulations could be improved, especially given that all private schools offered prohibited items. In addition, tables are informative but dense at the same time which may authors could consider highlighting key comparisons in bold or footnotes for easier interpretation. Around the references, they seem adequate but the Discussion section would benefit from more international comparisons (e.g. school food environments in other Latin American or countries with the similar income/condition). The policy section could also cite relevant WHO or FAO guidelines to reinforce the argument for local policy updates."
Response:
General Comments
We sincerely thank the reviewer for the careful reading and thoughtful feedback. We greatly appreciate your recognition of the relevance, structure, and methodological strengths of our study, including the stratification by school type and social vulnerability, the use of validated indicators, and the geospatial approach.
Below we address each of your suggestions point by point:
Comment 1: The authors should clarify the central research question earlier in the Introduction.
Response 1:
Thank you for this suggestion. We revised the final paragraph of the Introduction (lines 73–79) to explicitly state the central research question, which guides the study:
"How is the food environment configured inside and around public and private schools in the Federal District, and how is this configuration associated with social vulnerability?"
This clarification helps anchor the objectives and structure the narrative more clearly.
To address this gap, this study investigates the following research question: How is the food environment configured inside and around public and private schools in the DF, and how is this configuration associated with social vulnerability? To answer this question, we mapped the surroundings and characterized the internal food environment of public and private schools in the DF. We considered school type and the degree of social vulnerability in each region. This study provides novel, locally grounded evidence to guide more effective and context-sensitive school food policies in Brazil (lines 73 – 79).
Comment 2: More details on how the virtual audit was conducted and validated (e.g., explain the use of Google Street View) would increase reliability of the paper.
Response 2:
We agree and have expanded the description of the virtual audit process in the Methods section. We clarified that we used Google Street View to verify the existence, physical location, and signage of establishments listed in the administrative data (RAIS), correct inconsistencies in CNAE codes, and identify additional unlisted establishments visible in the surrounding area.
Establishments identified through RAIS were subsequently subjected to a virtual audit using Google Street View (2023). This process involved verifying the existence, geographic location, and building façade of each listed establishment. Appropriate corrections were applied when discrepancies were identified between the registered CNAE code and the actual business activity visible in the images. Additionally, other eligible establishments not originally listed were identified and included. Universal Transverse Mercator coordinates and corresponding zones were recorded for all verified establishments (lines 134–140).
Comment 3: Explain how potential selection bias from schools that declined participation was addressed.
Response 3:
We appreciate this important observation. In the revised Discussion section, we included a paragraph addressing the potential nonresponse bias caused by the refusal of two private schools to participate. Although the sample was probabilistic and stratified, we acknowledge that refusals may introduce some bias, especially if nonparticipating schools differed systematically from those included. This limitation is now discussed explicitly.
The reasons for refusal were not related to school characteristics. As in most probability samples, when there is no available data on non-respondents, we assumed that no systematic differences exist between participants and non-participants. This assumption minimized the risk of selection bias and supported the validity of the sample design. (lines 107 – 111)
Although two private schools declined participation, their refusal did not appear to be associated with any identifiable institutional characteristics. Nonetheless, as with most probability samples, it was not possible to compare respondents and non-respondents directly. Thus, we assumed that no systematic differences exist between these groups. While this assumption is methodologically acceptable, it introduces a potential, albeit limited, source of selection bias that should be acknowledged. (lines 534 – 539)
Comment 4: Policy implications could be better developed. It may also be worth discussing how enforcement of regulations could be improved, especially given that all private schools offered prohibited items.
Response 4:
Thank you for highlighting this point. We expanded the Discussion section to reflect on the barriers to policy implementation, particularly in private schools, and included suggestions for improving enforcement.
The implementation of school food regulations faces significant challenges, particularly in private institutions. These schools often show greater resistance to compliance, partly due to their limited integration into public food and nutrition policies, such as the National School Feeding Program (PNAE), which facilitates adherence in public schools [56,59]. The effectiveness of regulatory measures is further weakened by the lack of educational campaigns, limited intersectoral coordination, political and economic pressures from the private sector, and inadequate monitoring systems [59]. In the Federal District, although Law No. 5,321/2014 assigns the Health Department responsibility for inspecting school canteens [54], no systematic mechanisms for monitoring and enforcement have been effectively implemented. Although these barriers were not directly assessed in the present study, they are documented in the literature and provide important context for understanding the limited enforcement and compliance observed in some schools [56,59,60]. (lines 462 – 474)
Comment 5: Tables are informative but dense. Authors could consider highlighting key comparisons in bold or footnotes for easier interpretation.
Response 5: Thank you for this suggestion. We revised the tables to improve clarity and readability, particularly to support easier interpretation of the most relevant comparisons.
For example, in Table 1, we revised its structure and content to improve visual organization. Specifically, we removed the column “Schools with at least 1 establishment – N; %”, which concentrated contextual information that was better suited for narrative explanation. The most relevant findings from that column are now integrated into the text of the Results section, allowing for a more fluid and reader-friendly presentation. We also adjusted the column layout and labels to enhance alignment, consistency, and legibility.
These modifications aim to improve the overall presentation of the table and make the data more accessible to readers.
Comment 6: The Discussion section would benefit from more international comparisons (e.g., school food environments in other Latin American or countries with similar income/condition).
Response 6:
We fully agree and have expanded the Discussion with references to recent studies on school food environments in Chile, Mexico, Spain, and New York City, highlighting inequities in exposure to unhealthy food outlets based on school type and socioeconomic context. These additions help situate the findings in a broader international landscape and offer relevant parallels for Brazilian policy development.
International evidence, however, contrasts with the findings of the present study. Research conducted in cities such as Madrid, New York, Mexico City, and Santiago has shown that schools located in lower-income areas tend to be more exposed to unhealthy food retailers [43–46]. Barcelona, on the other hand, revealed a different pattern: although 90% of schools had at least two unhealthy food outlets nearby, those located in higher-income neighborhoods had significantly greater availability and affordability of healthy foods [47]. These findings highlight persistent socioeconomic disparities in school food environments across diverse global settings. Addressing such inequalities requires comprehensive and context-sensitive public policies that regulate the food environment both inside and around schools. (lines 390 – 400).
Several countries have also adopted regulations regarding the sale and advertising of food in school canteens, including Australia, Bulgaria, Chile, Canada, Costa Rica, South Korea, Ecuador, Estonia, France, Hungary, Mexico, Poland, and the United Kingdom. These policies aim to restrict the availability and marketing of unhealthy foods and promote healthier dietary practices among students. The diversity of national approaches highlights a global recognition of the school environment as a key setting for nutrition-related interventions and underscores the importance of regulatory frameworks to protect children’s health [48]. (lines 434 – 441)
Comment 7: The policy section could also cite relevant WHO or FAO guidelines to reinforce the argument for local policy updates.
Response 7:
Thank you for this excellent suggestion. We incorporated references to WHO and FAO recommendations on school food environments and the promotion of healthy eating among children and adolescents.
The World Health Organization (WHO) emphasizes the importance of mandating food and health education in the core school curriculum as a strategy to strengthen nutrition literacy and develop healthy eating skills among students, parents, and caregivers. However, to promote effective and lasting changes, educational efforts must be accompanied by structural interventions in the school food environment. In this regard, the WHO also recommends that governments establish clear nutritional standards for school meals and for foods sold within school premises, ensuring alignment with healthy eating guidelines. It further advises restricting the sale and marketing of unhealthy products in schools and creating buffer zones around them to limit children's exposure to obesogenic environments. These measures highlight the essential role of the education sector in addressing childhood obesity and fostering healthy habits from an early age [47].
4. Response to Comments on the Quality of English Language
The English in this article has been carefully revised with a focus on clarity, sentence structure, and academic style. The current version avoids long and complex sentences and ensures coherence across paragraphs. Furthermore, the manuscript has been professionally reviewed and proofread by a native English speaker with expertise in academic writing.